# Bi-Mamba: Towards Accurate 1-Bit State Space Models

**Shengkun Tang**[*]                                    *shengkun.tang@mbzuai.ac.ae*
*Department of Machine Learning*
*Mohamed bin Zayed University of Artificial Intelligence*

**Liqun Ma**[*]                                         *liqun.ma@mbzuai.ac.ae*
*Department of Machine Learning*
*Mohamed bin Zayed University of Artificial Intelligence*

**Haonan Li**                                           *haonan.li@mbzuai.ac.ae*
*Department of Natural Language Processing*
*Mohamed bin Zayed University of Artificial Intelligence*

**Mingjie Sun**                                         *mingjies@andrew.cmu.edu*
*Department of Computer Science*
*Carnegie Mellon University*

**Zhiqiang Shen**[†]                                    *zhiqiang.shen@mbzuai.ac.ae*
*Department of Machine Learning*
*Mohamed bin Zayed University of Artificial Intelligence*

**Reviewed on OpenReview:** *https://openreview.net/forum?id=CKQ4AgoRQm*

## Abstract

The typical Selective State-Space Model (SSM) used in Mamba addresses several limitations of Transformers, such as the quadratic computational complexity with respect to sequence length and the significant memory requirements during inference due to the key-value (KV) cache. However, the increasing size of Mamba models continues to pose challenges for training and deployment, particularly due to their substantial computational demands during both training and inference. In this work, we introduce `Bi-Mamba`, a scalable and powerful 1-bit Mamba architecture designed to enable more efficient large language models (LLMs), with model sizes of 780M, 1.3B, and 2.7B parameters. `Bi-Mamba` models are trained from scratch on a standard LLM-scale dataset using an autoregressive distillation loss. Extensive experiments on language modeling benchmarks demonstrate that `Bi-Mamba` achieves performance comparable to its full-precision (FP16 or BF16) counterparts, while outperforming post-training binarization (PTB) Mamba and binarization-aware training (BAT) Transformer baselines. Moreover, `Bi-Mamba` drastically reduces memory usage and computational cost compared to the original Mamba. Our work pioneers a new line of linear-complexity LLMs under low-bit representation and paves the way for the design of specialized hardware optimized for efficient 1-bit Mamba-based models. Code and the pre-trained weights are available at https://github.com/Tangshengku/Bi-Mamba.

## 1 Introduction

The Selective State-Space Model (SSM) (Gu et al., 2021b; 2022) has recently emerged as a powerful alternative to Transformers (Vaswani et al., 2017) in language modeling, achieving comparable or superior performance at small to moderate scales. SSMs scale linearly with sequence length during training and maintain a constant state size during generation, which significantly reduces computational and memory overhead, resulting in greater speed and efficiency.

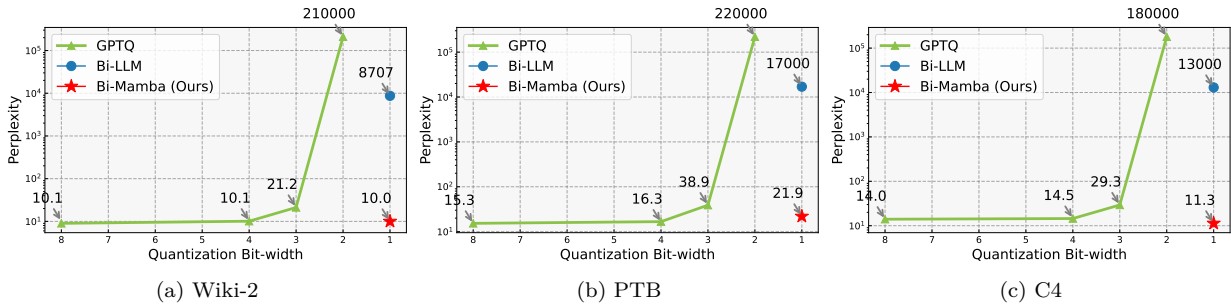

Figure 1: Perplexity comparison of `Bi-Mamba`, GPTQ and Bi-LLM on Wiki2, PTB and C4 datasets. GPTQ and Bi-LLM show significant performance degradation when the bit is low. `Bi-Mamba` demonstrates low perplexity in 1 bit and shows similar performance as GPTQ-8bit.

The representative SSM model, Mamba (Gu & Dao, 2024; Dao & Gu, 2024), has a clear advantage in handling long-context sequences due to its linear complexity. In contrast, conventional Transformers exhibit quadratic complexity as sequence length increases. This makes Mamba substantially more efficient for tasks involving large inputs or extended contexts, with memory and compute requirements that grow only linearly. In practice, this enables faster processing of long sequences with lower resource consumption, making Mamba well-suited for applications such as long-document processing, conversational agents, and other scenarios requiring large-context management. On the other hand, Transformers require polynomially more resources as context length grows, which quickly becomes a bottleneck in long-context tasks.

Prior work (Devlin et al., 2019; Radford et al., 2019; Touvron et al., 2023; Achiam et al., 2023) on Transformers has been extensive, with numerous methods proposed to reduce their high computational and storage costs, such as pruning (Ma et al., 2023), model quantization (Frantar et al., 2023), and KV cache optimizations (Pope et al., 2023; Kwon et al., 2023). Among these, quantization has proven highly effective (Dettmers et al., 2022b), enabling reductions from 16-bit to 8-bit, 4-bit, and even 1-bit representations with minimal performance loss (Frantar et al., 2023; Ma et al., 2024b;a).

However, little is known about how SSM models (Mamba in particular) perform under low-bit quantization or binarization. In this paper, we present `Bi-Mamba`, a novel approach that applies extreme quantization to SSM models by reducing weights to a binary setting. We show that SSM models can be effectively binarized for both training and inference, retaining high performance while dramatically reducing memory footprint and energy consumption.

Our work pioneers a new framework for 1-bit representation in linear computational complexity models, potentially facilitating the development of specialized hardware tailored for fully low-bit SSM-based language models. We provide comprehensive experiments showing that `Bi-Mamba` achieves competitive performance compared to its full-precision counterpart large language models (LLMs), thereby establishing the feasibility and benefits of binarizing SSM models. To further understand the behaviors of binarization on Mambas, we conducted extensive empirical studies to analyze the distribution of pre-trained and binarized weights in Mambas. The results of this study are detailed in Section 4 and Appendix A, leading to two key observations:

- As shown in Figure 2, post-training-binarization methods like BiLLM (Huang et al., 2024) and PB-LLM (Shang et al., 2023) typically tend to shift the distribution of weights on Mamba after binarization, resulting in a misaligned distribution to the optimal binary weights. Our binarization-aware training, however, ensures that the binarized weights remain close to the original weight distribution, preserving the largest capability in weight representation binarization.

- Based on our empirical experiments, applying existing LLM post-binarization methods (Frantar et al., 2023; Huang et al., 2024), even when retaining salient weights, often severely degrades the performance of the Mamba model. Without accounting for salient weights, binarization-aware training appears to be the only feasible solution for effectively binarizing models like Mamba while preserving competitive performance.

---

*Equal contribution. †Corresponding author.

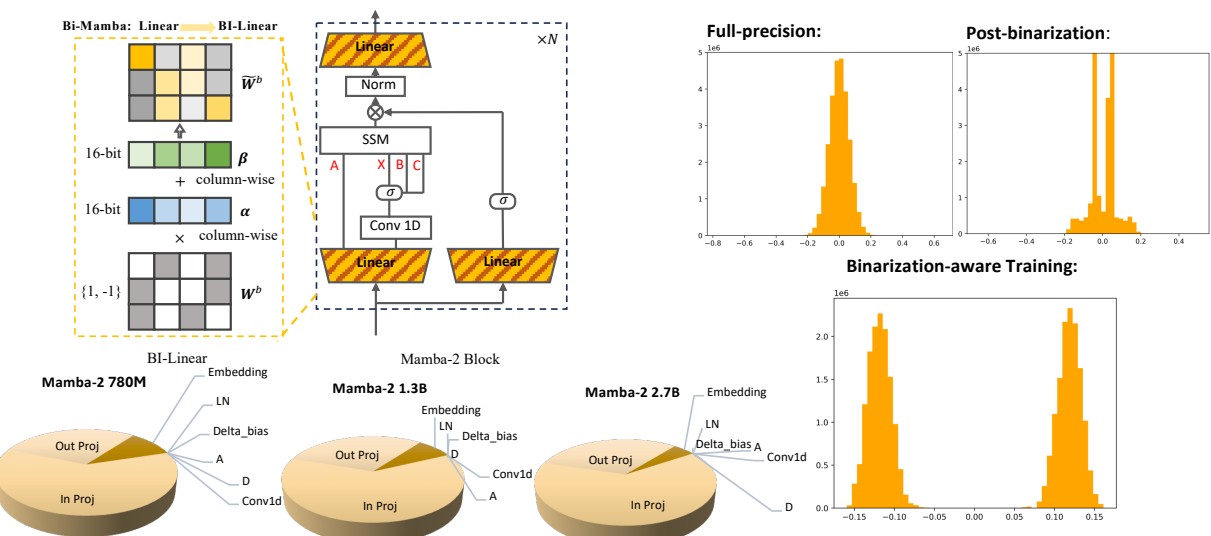

Figure 2: Illustration of the `Bi-Mamba` framework. Our `Bi-Mamba` binarizes both input and output projection matrices. Compared with the post-binarization method (Bi-LLM), our binarization-aware training method (`Bi-Mamba`) generates a more similar weight distribution (after scaling) on each part.

Our contributions in this paper are as follows: (1) This is the first work to successfully binarize the Mamba architecture to 1-bit from scratch while maintaining strong performance. (2) We explore the potential parameter space for binarization in Mamba, offering valuable insights for future research. (3) The `Bi-Mamba` model pretrained by our method can serve as a robust base model for downstream tasks in resource-limited scenarios and can be easily adapted for other NLP applications.[*]

## 2 Related Work

**Post-Training Quantization.** Due to their large parameter count, LLMs (Brown et al., 2020; OpenAI, 2023; Touvron et al., 2023) are resource intensive to run. Therefore, they are often quantized to low-bit representations during inference. This type of quantization is typically called post-training quantization (PTQ) (Dettmers et al., 2022a; Xiao et al., 2023; Wei et al., 2022; Frantar et al., 2023; Yao et al., 2022; Xiao et al., 2023; Lin et al., 2024), where the quantization operation is applied to the pretrained models. Post-training quantization methods of LLMs are typically based on the empirical observations that a small set of salient features in pretrained Transformers have significantly large magnitudes, and quantization of these features needs to be extra careful (Xiao et al., 2023; Lin et al., 2024). PTQ methods have achieved nearly lossless accuracy for INT8 weights-and-activations quantization and INT4 weight-only quantization. While many works (Huang et al., 2024; Egiazarian et al., 2024; Shao et al., 2023; Chee et al., 2023; Tseng et al., 2024) have focused on pushing post-training weight quantization below 4 bits, they often lead to a drastic drop in model performance.

**Quantization-Aware Training.** Different from post-training quantization, quantization-aware training (QAT) aims to learn the quantized models during training. In the LLM era, QAT is less investigated as compared to PTQ because LLMs typically require huge amounts of compute resources to train. Li et al. (2023b) propose a method that adopts LoRA fine-tuning during QAT, making the quantization procedure more resource-efficient. However, these methods still require the pretrained weights to initialize the student LLMs. Wortsman et al. (2023) propose a method to stabilize the training of low-bit large vision-language models from random initializations. Most recently, Ma et al. (2024b) trained LLMs with ternary weights from random initializations, achieving non-trivial performance. Ma et al. (2024a) further proposed a distillation based method to pretrain binarized LLMs from scratch.

---

[*]We will make all our models, code, and training datasets fully available, we aim to support further research in this direction and encourage the exploration of binarized SSM models for more efficient large language models.

| Model Size | Embedding | LN | $\Delta_{bias}$ | A | D | Conv1d | In Proj. | Out Proj. |
|---|---|---|---|---|---|---|---|---|
| Mamba-2 780M | 9.901 | 0.029 | 0.0003 | 0.0003 | 0.0003 | 0.102 | 60.936 | 29.031 |
| Mamba-2 1.3B | 7.664 | 0.022 | 0.0002 | 0.0002 | 0.0002 | 0.078 | 62.270 | 29.964 |
| Mamba-2 2.7B | 4.763 | 0.018 | 0.0002 | 0.0002 | 0.0002 | 0.064 | 64.115 | 31.039 |

Table 1: Proportional distribution of parameters across different modules in Mamba-2. Input and output matrices take up the majority of the parameters in Mamba-2 models, around 90% and more.

**State-Space Models.** While LLMs are often built with Transformers (Vaswani et al., 2017), their self-attention operations suffer from quadratic time complexity. This makes Transformers inefficient to run on long sequences. Therefore there have been many research efforts aiming at addressing the efficiency issue (Peng et al., 2023; Beck et al., 2024; Katharopoulos et al., 2020; De et al., 2024). Among these efforts, SSMs (Gu & Dao, 2024; Dao & Gu, 2024; Gu et al., 2021a) are a type of recurrent neural networks. The latest advanced SSMs like Mamba (Gu & Dao, 2024) and Mamba-2 (Dao & Gu, 2024) have demonstrated comparable performance to Transformers, while having linear complexity with respect to the sequence length. Despite their promising capabilities, how to effectively quantize this architecture has rarely been investigated. Concurrent to our work, Pierro & Abreu (2024) have looked at post-training quantization of Mamba.

## 3 Approach

### 3.1 Preliminary: Mamba Series

Mamba belongs to a class of models known as State Space Models (SSMs), which is able to offer performance and scaling laws comparable to the Transformer while remaining practical at extremely long sequence lengths (e.g., one million tokens). It achieves this extended context by eliminating "quadratic bottleneck" present in the attention mechanism. The vanilla structured state space sequence models (S4) transform a 1-dimensional sequence $x \in R^T$ into another sequence $y \in R^T$ by utilizing a latent state $h \in R^{T,N}$ as follows:

$$\dot{h_t} = Ah_t + Bx_t$$
$$y_t = Ch_t + Dx_t \tag{1}$$

where $A \in R^{N,N}$, $B \in R^{N,1}$, $C \in R^{1,N}$, $D \in R$. $h_{t-1}$ is the hidden state, $x_t$ is the input, the observation that the model gets each time. $h_t$ then represents the derivative of the hidden state, i.e. how the state is evolving.

Since the parameters in Equation 1 are constant through time, this model is linear time-invariant (LTI). The LTI model gives equal attention to all elements when processing sequences, which prevents the model from effectively understanding natural language. To address this, Mamba improved it to the Selective State Space Model, where $B$ and $C$ are obtained through a linear projection of $x_t$. Meanwhile, the above equation applies to dynamic systems with continuous input and output signals. So, it needs to do discretization when dealing with text sequences:

$$\overline{A} = exp(\Delta A)$$
$$\overline{B} = exp(\Delta A)^{-1}(exp(\Delta A) - I) \cdot \Delta B \tag{2}$$

where $\Delta$ depends on $x_t$ and a learnable parameter $\Delta_{bias}$. Then, the calculating process of selective SSM is as follows:

$$h_t = \overline{A}h_{t-1} + \overline{B}x_t$$
$$y_t = Ch_t + Dx_t \tag{3}$$

Centered around selective SSM, combined with linear projection for input and output along with layer normalization, this forms the basic block for Mamba.

Compared to Mamba, Mamba-2 further replaces selective SSM with the state space duality (SSD). In SSD, $A$ is simplified as scalar-times-identity structure, and SSD uses a larger head dimension, which is 1 in Mamba. Meanwhile, Mamba-2 block also introduces simplifications, such as removing sequential linear projections in the Mamba block, to facilitate parallel training. The basic structure of Mamba-2 is shown in Figure 2.

### 3.2 Bi-Mamba

**Binarization Space in Mamba.** To begin with, we need to identify what weight matrices can be binarized in the Mamba architecture. We use the latest Mamba-2 (Dao & Gu, 2024) as our base architecture. According to the description above, we can interpret the SSD matrices of $A$, $B$, $C$, $D$, and $\Delta_{bias}$ more intuitively, as well as other layers including *embedding, layer normalization (LN), Conv-1d, inner linear projection, out linear projection* in Mamba-2 to better understand the effect after binarization, so that to determine the binarization space. Briefly, $A$ is the transition state matrix, representing how the current state transitions to the next state. It answers *how should the model gradually forget the less relevant parts of the state over time? B* maps the new input to the state, addressing the point of *which parts of the new input should the model retain? C* maps the state to the output of the SSM, asking *how can the model utilize the current state to make an accurate next prediction? D* shows how the new input directly influences the output, acting as a modified skip connection and asking *how can the model incorporate the new input into the prediction?*

Considering our base architecture Mamba-2, we present the specific parameters proportion for different layers across different sizes as shown in Figure 2 and Table 1[*]. It is observed that, in Mamba-2, the vast majority of the model's parameters are in the linear modules, excluding the causal head. For instance, in Mamba-2-2.7B, these parameters account for 95.2% of the entire model. Additionally, the embedding module shares parameters with the causal head. Binarizing the embedding significantly diminishes its capability to represent token semantics, thereby reducing model performance. Therefore, in our Bi-Mamba, we only binarize the parameters in the linear modules (excluding the causal head). This strategy aims to maintain a high compression ratio of 90% while still allowing the binarized model to perform effectively.

**Simple Learnable Scaling Factors for Better Capability.** To binarize Mamba, we replace the original linear modules with the FBI-Linear module introduced by FBI-LLM (Ma et al., 2024a). We note that our primary goal is to demonstrate the feasibility of binarizing the Mamba architecture and show that it can achieve strong performance, not to claim that FBI-LLM is the optimal scheme here. While leveraging this scheme, our core motivation is the successful adaptation of binarization-aware training to the distinct SSM architecture of Mamba. Specifically, the FBI-Linear module primarily consists of two parts: a matrix $\boldsymbol{W}^b \in \mathbb{R}^{m \times n}$ made up solely of $\{1, -1\}$ and high-precision scale factors $\boldsymbol{\alpha} \in \mathbb{R}^n$ and $\boldsymbol{\beta} \in \mathbb{R}^n$. The inference process of FBI-Linear is as follows:

$$\boldsymbol{y} = \widetilde{\boldsymbol{W}}^b \boldsymbol{x} \tag{4}$$

where $\widetilde{\boldsymbol{W}}^b$ is derived by performing column-wise multiplication with $\boldsymbol{\alpha}$ and addition with $\boldsymbol{\beta}$ respectively:

$$\widetilde{\boldsymbol{W}}^b_{\cdot,i} = \alpha_i \boldsymbol{W}^b_{\cdot,i} + \beta_i \tag{5}$$

where $\alpha_i$ and $\beta_i$ are the learnable scaling and shifting factors at $i$-th layer.

**Objective of Training.** Our training objective is a cross-entropy loss between the outputs of the target student model and the pretrained teacher model at each step of the autoregressive scheme for next-token prediction. This can be expressed as:

$$\mathcal{L}_{\texttt{Bi-Mamba}} = -\frac{1}{n} \sum_{k}^{n} \boldsymbol{p}^{\mathcal{T}}\left(x^{k+1}\right) \cdot \log \boldsymbol{p}^{\mathcal{S}}\left(x^{k+1}\right) \tag{6}$$

where $n$ is the number of input tokens. Here, $\boldsymbol{p}^{\mathcal{T}}\left(x^{k+1}\right)$ represents the token distribution over the vocabulary at the $k$-th step predicted by the teacher model, and $\boldsymbol{p}^{\mathcal{S}}\left(x^{k+1}\right)$ is the corresponding predicted distribution by the student model.

**Overall Design of Bi-Mamba.** During binarization-aware training of autoregressive distillation (Ma et al., 2024a), we compute the cross-entropy between the output probability distributions of a high-precision pretrained model and our target Bi-Mamba model. In this process, $\boldsymbol{\alpha}$ and $\boldsymbol{\beta}$ are learnable parameters, while $\boldsymbol{W}^b$ is derived using the $sign(\cdot)$ function from a learnable high-precision matrix $\boldsymbol{W}^f \in \mathbb{R}^{m \times n}$. Since the $sign(\cdot)$ function is non-differentiable, we use the Straight Through Estimator (STE) (Bengio et al., 2013) as prior

---

[*]Since $B$ and $C$ are derived from the linear projection of each layer's inputs, and the language model head is tied to the embedding layer, these modules are ignored in our design.

studies (Rastegari et al., 2016; Alizadeh et al., 2019) in BNNs to approximate the gradients of the input variables, enabling the continuation of backward propagation.

**Complexity Analysis.** Let $L$ denote the sequence length, $d$ the model embedding width, and $r \times d$ the hidden width of the MLP (usually $r \approx 4$), the Transformer cost is $\mathcal{O}(L^2 d)$ for self-attention, $\mathcal{O}(Ld^2)$ to form $Q$, $K$, $V$ plus $\mathcal{O}(L^2 d)$ for the $QK^\top$ product, and $\mathcal{O}(L \times d^2)$ for MLP. For Mamba, because each token is processed by a single state-space update, the per-layer complexity is $\mathcal{O}(L \times d)$ and each token is one SSM step. For the binary variants, Binary Transformer has the same algebraic graph $\implies O(L^2 \times d)$ bit-ops, but a multiply-accumulate is replaced by a single binary operator like XNOR and popcount on 32- or 64-bit elements, yielding $\approx 32\times$ fewer loaded bits in weight bandwidth. Our Bi-Mamba also has $\mathcal{O}(L \times d)$ bit-ops (inheriting Mamba's linear scaling) and achieves a comparable $\approx 32\times$ reduction in weight bits.

## 4 Experiments

In our experiments, we solely binarize the parameters in the most linear modules, while keeping other model parameters and activations at their original precision. Notably, we do not strictly represent the binarized parameters with 1-bit, instead, we use high-precision values to simulate the binarized parameters. We train `Bi-Mamba` on different scales and evaluate their performance across multiple tasks.

### 4.1 Setup

**Training Dataset.** Following FBI-LLM, we train `Bi-Mamba` with the Amber dataset (Liu et al., 2023) which contains a total 1.26 Trillion tokens from RefinedWeb (Penedo et al., 2023), StarCoder (Li et al., 2023a), and RedPajama-v1 (Computer, 2023). The data is partitioned into 360 chunks, each comprising approximately 3.5B tokens on average.

**Training details.** We train `Bi-Mamba` on different scales with Mamba-2 architecture. Specifically, we binarize input projection and output projection matrices in 780M, 1.3B and 2.7B Mamba-2 models. We use LLaMA2-7B as the teacher model for all `Bi-Mambas` to calculate autoregressive distillation loss. Therefore, all `Bi-Mambas` we trained have the same vocabulary and tokenizer as LLaMA2. We train models until convergence with $32\times$ NVIDIA A100 GPUs in total and maintain BF16 precision while training. For configuring different sizes of `Bi-Mamba`, the details can be found at Table 2. We follow the same architectures as the original Mamba-2 models and apply binarization on both input and output projection matrices. The training process uses the Adam optimizer with parameters $\beta_1 = 0.9$ and $\beta_2 = 0.95$. The initial learning rate is set at $2.5e^{-4}$ and follows a cosine schedule, decreasing to $2.5e^{-5}$ over 2,000 warm-up steps. Gradient clipping is set at 1.0. We train `Bi-Mamba` 780M, 1.3B, 2.7B with 30 data chunks, which are 105B tokens.

|  | Bi-Mamba 780M | Bi-Mamba 1.3B | Bi-Mamba 2.7B |
|---|---|---|---|
| d_model | 1536 | 2,048 | 2,560 |
| n_layer | 48 | 48 | 64 |
| vocabulary size | 32,000 | 32,000 | 32,000 |
| learning rate | $2.5e^{-4}$ | $2.5e^{-4}$ | $2.5e^{-4}$ |
| batch size (token) | 0.5M | 0.5M | 0.5M |
| teacher model | LLaMA2-7B | LLaMA2-7B | LLaMA2-7B |

Table 2: The configuration and training details for `Bi-Mamba`.

**Evaluation Metrics.** We evaluate the models based on their zero-shot performance in downstream tasks, including BoolQ (Clark et al., 2019), PIQA (Bisk et al., 2020), HellaSwag (Zellers et al., 2019), WinoGrande (Sakaguchi et al., 2021), ARC (Clark et al., 2018), and OpenbookQA (Mihaylov et al., 2018). We also evaluate `Bi-Mamba` and other baselines on HumanEval and GSM8K datasets. All downstream evaluations are done with *lm-evaluation-harness* package (Gao et al., 2024). We also use perplexity on Wikitext2 (Merity et al., 2016), PTB (Marcus et al., 1993), C4 (Raffel et al., 2020) dataset as the evaluation metric. Perplexity measures how well a probability model predicts a token, quantitatively measuring the model's generation power.

**Baselines.** We compare our work with quantization and binarization methods, namely GPTQ (Frantar et al., 2023), SqueezeLLM (Kim et al., 2023), AQLM (Egiazarian et al., 2024), and Bi-LLM (Huang et al.,

| Method | Model | Size | Zero-shot Accuracy ↑ | | | | | | | | Perplexity ↓ | | |
|---|---|---|---|---|---|---|---|---|---|---|---|---|---|
| | | | BoolQ | PIQA | HS | WG | ARC-e | ARC-c | OBQA | Avg. | Wiki2 | PTB | C4 |
| Mamba-2 (Dao & Gu, 2024) | M | 780M | 61.5 | 71.8 | 54.9 | 60.2 | 54.3 | 28.5 | 36.2 | 52.5 | 11.8 | 20.0 | 16.5 |
| GPTQ-3bit | M | 780M | 44.6 | 62.9 | 40.3 | 53.3 | 40.6 | 26.4 | 30.6 | 42.6 | 152.5 | 192.5 | 186.0 |
| SqueezeLLM-3bit | M | 780M | 61.4 | 69.7 | 50.6 | 56.2 | 51.3 | 27.6 | 30.2 | 49.6 | 15.5 | 25.3 | 21.2 |
| GPTQ-2bit | M | 780M | 40.4 | 52.3 | 25.7 | 51.3 | 25.6 | 25.1 | 30.2 | 35.2 | 1.6e+8 | 1.3e+8 | 7.3e+7 |
| SqueezeLLM-2bit | M | 780M | 40.3 | 58.3 | 33.6 | 51.5 | 38.0 | 24.7 | 27.0 | 39.1 | 141.7 | 216.3 | 323.4 |
| AQLM-2.04bit | M | 780M | 38.8 | 57.3 | 30.0 | 50.1 | 33.1 | 22.4 | 25.4 | 36.7 | 245.2 | 332.4 | 311.0 |
| BiLLM | M | 780M | 54.1 | 52.9 | 26.9 | 50.6 | 28.5 | 26.5 | 27.2 | 38.1 | 1.8e+4 | 2.4e+4 | 1.5e+4 |
| BitNet-1.58bit† | T | 700M | 58.2 | 68.1 | 35.1 | 55.2 | 51.8 | 21.4 | 20.0 | 44.3 | - | - | - |
| Onebit* | M | 780M | 62.2 | 63.3 | 38.6 | 54.4 | 44.6 | 24.4 | 28.6 | 45.1 | 26.3 | 39.6 | 27.4 |
| Bi-Mamba | M | 780M | 58.5±0.039 | 68.0±0.471 | 41.6±0.0774 | 52.0±0.035 | 42.4±0.084 | 24.3±0.106 | 30.6±0.459 | **45.3**±0.162 | **13.4**±0.124 | **32.4**±0.0959 | **14.5**±0.116 |
| TinyLLaMA (Zhang et al., 2024) | T | 1.3B | 57.8 | 73.3 | 59.2 | 59.1 | 55.3 | 30.1 | 36.0 | 53.0 | 7.8 | 30.5 | 9.9 |
| OPT (Zhang et al., 2022) | T | 1.3B | 57.8 | 72.5 | 53.7 | 59.5 | 51.0 | 29.5 | 33.4 | 51.1 | 14.6 | 20.3 | 16.1 |
| Mamba-2 (Dao & Gu, 2024) | M | 1.3B | 64.3 | 73.7 | 59.9 | 61.0 | 60.4 | 33.1 | 37.8 | 55.8 | 10.4 | 17.7 | 14.8 |
| GPTQ-3bit | M | 1.3B | 56.8 | 68.2 | 48.5 | 54.4 | 48.0 | 28.8 | 30.4 | 47.8 | 29.3 | 56.5 | 37.3 |
| SqueezeLLM-3bit | M | 1.3B | 62.9 | 72.2 | 56.6 | 58.6 | 59.0 | 32.3 | 35.8 | 53.9 | 11.8 | 20.4 | 16.7 |
| GPTQ-2bit | M | 1.3B | 42.0 | 49.9 | 25.7 | 49.6 | 26.4 | 26.1 | 27.6 | 35.3 | 1.2e+6 | 1.0e+6 | 1.3e+6 |
| SqueezeLLM-2bit | M | 1.3B | 62.3 | 64.3 | 41.3 | 53.9 | 47.5 | 26.2 | 30.4 | 46.5 | 31.9 | 300.2 | 136.4 |
| AQLM-1.92bit | M | 1.3B | 47.1 | 57.5 | 33.9 | 51.3 | 36.7 | 23.2 | 27.6 | 39.6 | 179.0 | 284.5 | 219.7 |
| BiLLM | M | 1.3B | 40.1 | 55.4 | 29.6 | 50.7 | 30.6 | 21.8 | 25.4 | 36.2 | 4943.2 | 3540.8 | 4013.6 |
| BitNet-1.58bit† | T | 1.3B | 56.7 | 68.8 | 37.7 | 55.8 | 54.9 | 24.2 | 19.6 | 45.4 | - | - | - |
| OneBit* | M | 1.3B | 61.6 | 65.3 | 43.1 | 55.1 | 47.3 | 25.0 | 30.6 | 46.8 | 20.9 | 30.9 | 23.4 |
| FBI-LLM† | T | 1.3B | 60.3 | 69.0 | 42.3 | 54.0 | 43.6 | 25.3 | 29.6 | 46.3 | 12.6 | 39.3 | 13.8 |
| Bi-Mamba | M | 1.3B | 60.0±0.025 | 68.8±0.435 | 47.3±0.021 | 55.9±0.0299 | 48.0±0.090 | 26.3±0.102 | 32.2±0.474 | **48.4**±0.167 | **11.7**±0.111 | **29.9**±0.116 | **12.9**±0.105 |
| Mamba-2 (Dao & Gu, 2024) | M | 2.7B | 70.7 | 76.3 | 66.6 | 63.9 | 64.8 | 36.3 | 38.8 | 59.6 | 9.1 | 15.3 | 13.3 |
| GPTQ-3bit | M | 2.7B | 54.8 | 69.9 | 54.0 | 56.0 | 51.6 | 33.3 | 32.8 | 50.3 | 21.2 | 39.0 | 29.3 |
| SqueezeLLM-3bit | M | 2.7B | 68.3 | 74.6 | 63.0 | 62.9 | 61.9 | 34.3 | 39.2 | 57.7 | 10.8 | 18.2 | 15.4 |
| GPTQ-2bit | M | 2.7B | 45.4 | 49.8 | 25.8 | 52.0 | 25.8 | 25.8 | 26.0 | 35.8 | 2.1e+5 | 2.3e+5 | 1.8e+5 |
| SqueezeLLM-2bit | M | 2.7B | 47.0 | 49.6 | 26.0 | 48.4 | 26.2 | 24.8 | 26.6 | 35.5 | 1.3e+5 | 3.2e+4 | 1.7e+5 |
| AQLM-2.09bit | M | 2.7B | 57.1 | 64.7 | 42.6 | 53.4 | 45.2 | 25.7 | 27.6 | 45.2 | 31.3 | 55.4 | 45.8 |
| BiLLM | M | 2.7B | 52.8 | 53.8 | 27.7 | 53.0 | 29.1 | 25.1 | 28.2 | 38.5 | 8707.0 | 1.7e+4 | 1.3e+4 |
| OneBit* | T | 6.7B | 63.3 | 67.7 | 52.5 | 58.1 | 41.6 | 29.3 | 34.0 | 49.5 | 10.2 | **18.2** | 11.6 |
| BitNet-1.58bit† | T | 3.0B | 61.5 | 71.5 | 42.9 | 59.3 | 61.4 | 28.3 | 26.6 | 50.2 | - | - | - |
| OneBit* | M | 2.7B | 60.3 | 68.9 | 50.3 | 60.5 | 49.5 | 30.0 | 33.0 | 50.3 | 16.5 | 24.7 | 19.8 |
| Bi-Mamba | M | 2.7B | 58.0±0.0111 | 72.5±0.448 | 54.3±0.035 | 56.1±0.068 | 51.4±0.104 | 29.1±0.112 | 32.6±0.493 | **50.6**±0.171 | **10.0**±0.105 | 21.9±0.114 | **11.3**±0.121 |

Table 3: Performance comparison with baselines on downstream tasks and perplexity. Here, *Model* represents the architecture of the quantized model. We divide the table into three blocks based on model size. Our Bi-Mamba achieves lower perplexity than Bi-LLM and GPTQ on Wiki2, PTB and C4 datasets, as well as the best average performance on downstream tasks compared with GPTQ-2bit and Bi-LLM. † refers to the results are from the original paper while * means the results are produced by the original implementation and weights. All other methods are implemented by us and applied on Mamba architecture. The results of Bi-Mamba are reported with standard deviation over 10 runs, each using different subsets of the test data.

2024). GPTQ, SqueezeLLM, and AQLM are post-training quantization methods, whereas Bi-LLM is a post-training binarization method. We apply the official implementation of GPTQ and SqueezeLLM and quantize the Mamba-2 models into 3 and 2 bits, respectively. For AQLM, we use their official implementation to quantize the 780M, 1.3B and 2.7B models into 2.04bit, 1.92bit and 2.09bit. For Bi-LLM, we also utilize their official implementation and binarize Mamba-2 models. Moreover, we add quantization-aware training methods, OneBit (Xu et al., 2024) and BitNet-1.58bit (Ma et al., 2024b) for comparison. For BitNet, we report the results from the original paper for comparison, whereas for OneBit, since only the official 7B weights were released, we include results only for the 7B models. We also add the comparison with FBI-LLM (Ma et al., 2024a), which is a transformer-based binary model. We report the results of FBI-LLM 1.3B model from their paper. Furthermore, we include results from open-source full-precision transformer-based models of various sizes, such as OPT (Zhang et al., 2022), and TinyLLaMA (Zhang et al., 2024), as references.

## 4.2 Main Results

Table 3 * presents the comparison of our Bi-Mamba model against various baselines on downstream tasks and perplexity on Wiki2, PTB, and C4 datasets. These evaluations provide insight into model generalization capabilities without further task-specific fine-tuning. The visualization of performance comparison is shown in Figure 3. The performance on HumanEval and GSM8K is presented in Table 4.

For the 780M Mamba-2 model, Bi-Mamba demonstrates an average downstream performance of 45.3, outperforming GPTQ-3bit and Bi-LLM, which achieve 42.6 and 38.1, respectively. In perplexity assessments, Bi-Mamba reports scores of 13.4, 32.4, and 14.5 on Wiki2, PTB, and C4, respectively, with the baseline models exhibiting up to 10× higher perplexity. SqueezeLLM and AQLM achieve better performance than

---

*In the table, "M" indicating Mamba-2 (Dao & Gu, 2024) and "T" indicating Transformer (Vaswani et al., 2017). *HS*, *WG*, and *OBQA* are abbreviations for HellaSwag, WinoGrande, and OpenbookQA, respectively.

GPTQ on both downstream tasks and perplexity, yet they are still outperformed by the `Bi-Mamba` 780M model. Moreover, `Bi-Mamba` 780M surpasses the binarization-aware training method, BitNet in the zero-shot performance on downstream tasks. BitNet obtains 44.3 scores on average while `Bi-Mamba` 780M achieves 45.3 scores.

For the 1.3B model, `Bi-Mamba` achieves a notable downstream accuracy of 48.4 on average, surpassing GPTQ-2bit's 35.3, SqueezeLLM-2bit's 46.5, AQLM's 39.6 and Bi-LLM's 36.2. This performance indicates `Bi-Mamba`'s enhanced generalization across a wider range of tasks at this model size. Additionally, `Bi-Mamba` demonstrates substantially improved perplexity, registering scores of 11.7, 29.9, and 12.9 on Wiki2, PTB, and C4 datasets, respectively. In comparison, GPTQ-2bit and Bi-LLM present considerably higher perplexity values, underscoring the efficiency of `Bi-Mamba`'s binarization in maintaining linguistic coherence. Moreover, `Bi-Mamba` 1.3B model beats the training-based method, BitNet and FBI-LLM, which obtains 45.4, 46.3 scores on average on the downstream tasks.

For the 2.7B model, `Bi-Mamba` further extends its lead, achieving an average downstream accuracy of 50.6, compared to 35.8 for GPTQ-2bit, 35.5 for SqueezeLLM-2bit, 45.2 for AQLM-2.09bit, and 38.5 for Bi-LLM. Notably, `Bi-Mamba` maintains low perplexity across all datasets, with scores of 10.0, 21.9, and 11.3 on Wiki2, PTB, and C4, respectively. These results highlight `Bi-Mamba`'s ability to retain high-level performance in both task accuracy and linguistic fluency as model complexity scales up.

While BitNet-1.58bit does not provide each perplexity respectively, the average perplexity of the 3B model is 9.91 on C4 and Wikitext2, which is reported in their original paper with 8-bit activation and 1.58-bit weights. In contrast, `Bi-Mamba` keeps 16-bit activation and binarizes weights, obtaining 10.65 on average. BitNet-1.58b shows slightly better perplexity, which we attribute to its larger size—3B parameters compared to Bi-Mamba's 2.7B—and its higher precision at 1.58 bits. Moreover, OneBit on the transformer model only obtains 49.5 scores on average with 6.7B parameters, and BitNet gains 50.2 scores with 3.0B parameters. On the Mamba 780M with the 1.3B model, Bi-Mamba achieves better performance in both perplexity and downstream tasks; however, on the 2.7B model, Bi-Mamba shows improved perplexity but obtains comparable downstream task performance to Onebit. Bi-Mamba's lower perplexity shows stronger language modeling, but this does not always yield higher task accuracy—especially when datasets' knowledge coverage limits model capability.

In summary, `Bi-Mamba` consistently demonstrates superior zero-shot performance and perplexity reduction across model sizes, substantiating its robustness and versatility compared to GPTQ and Bi-LLM, particularly in larger, more complex models.

For results on HumanEval and GSM8K, we notice that the full-precision models across all sizes perform poorly, producing results close to random guessing. Other quantization methods further degrade performance compared to their full-precision counterparts. In contrast, our proposed Bi-Mamba consistently achieves superior performance, outperforming all other methods, including the full-precision model. We believe the reason is the higher quality of pretraining data. Bi-Mamba applied Amber dataset, which is a high-quality dataset compared with Pile dataset used in the original Mamba model. We further conduct the instruction tuning on our base Bi-Mamba model with OpenOrca (Lian et al., 2023) dataset. We trained it for 3.3B tokens. After instruction-tuning, the performance is further improved, demonstrating that Bi-Mamba can serve as a strong base model. Overall, the results of the Mamba models and their quantized versions are not highly competitive. The primary reasons for this are: 1) Limited Training Data. The Mamba-2-2.7B pre-trained model is trained with only 300B tokens on the Pile dataset. This amount of training data is relatively small compared to state-of-the-art models, which are often trained on significantly larger corpora (more than 10T tokens). As a result, the pre-trained model may not have fully developed strong reasoning and problem-solving capabilities. After the quantization or binarization, the reasoning ability collapses completely. 2) Lack of Instruction Tuning. The models in this evaluation have not been instruction-tuned, which is critical for performance on complex reasoning and coding benchmarks such as GSM8K and HumanEval. Instruction tuning enhances a model's ability to follow prompts effectively and generalize across tasks. Quantizing an instructed model could yield better performance on complex reasoning tasks.

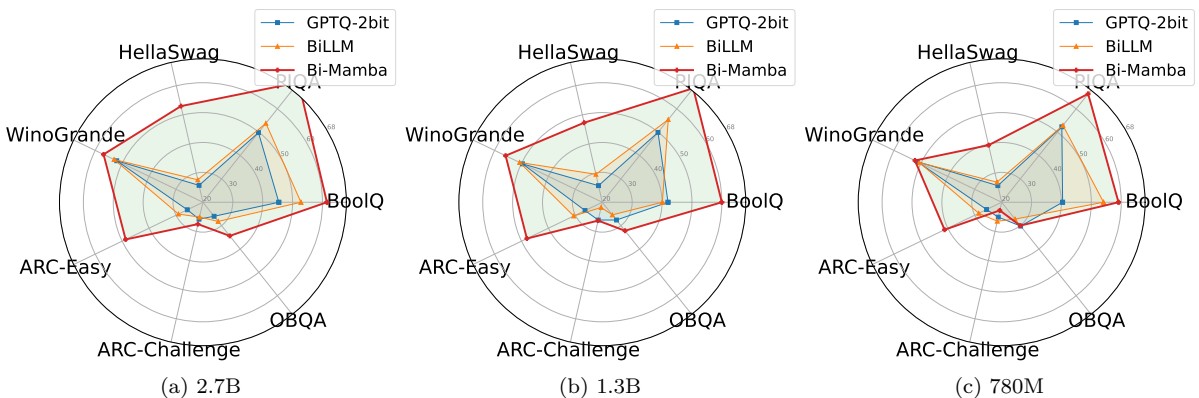

Figure 3: Visualization of results comparison on Mamba-2 in the scales of 2.7B, 1.3B and 780M.

| Model | Model Param. | HumanEval | GSM8K |
|---|---|---|---|
| Mamba-2-16bit | 780M | 0.81 | 0.76 |
| GPTQ-2bit | 780M | 0.00 | 0.68 |
| AQLM-2.04bit | 780M | 0.00 | 0.61 |
| SqueezeLLM-2bit | 780M | 0.00 | 0.00 |
| BiLLM-2bit | 780M | 0.00 | 0.07 |
| Bi-Mamba | 780M | **0.85** | **1.29** |
| Bi-Mamba (Instruct) | 780M | **1.22** | **1.97** |
| Mamba-2-16bit | 1.3B | 1.83 | 0.76 |
| GPTQ-2bit | 1.3B | 0.23 | 0.37 |
| AQLM-1.92bit | 1.3B | 0.20 | 0.38 |
| SqueezeLLM-2bit | 1.3B | 0.20 | 0.15 |
| BiLLM-2bit | 1.3B | 0.18 | 0.15 |
| Bi-Mamba | 1.3B | **0.71** | **1.95** |
| Bi-Mamba (Instruct) | 1.3B | **3.66** | **2.88** |
| Mamba-2-16bit | 2.7B | 1.07 | 0.91 |
| GPTQ-2bit | 2.7B | 0.00 | 0.68 |
| AQLM-2.09bit | 2.7B | 0.26 | 0.53 |
| SqueezeLLM-2bit | 2.7B | 0.23 | 0.23 |
| BiLLM-2bit | 2.7B | 0.30 | 0.38 |
| Bi-Mamba | 2.7B | **0.91** | **1.29** |
| Bi-Mamba (Instruct) | 2.7B | **6.10** | **3.94** |

Table 4: Performance comparison of different methods on HumanEval and GSM8K evaluation set.

## 5 Analysis

### 5.1 Training Result Dynamics

In this section, we discuss the performance of `Bi-Mamba` as the training progresses with different training costs/budgets. The main results are shown in Figure 4. We present the downstream performance and perplexity curves across different training costs for the Mamba-2-780M model. More results on 2.7B and 1.3B Mamba-2 models can be found in the Appendix A.1. From the figure, we can observe that the perplexity decreases quickly at the beginning of training and gradually converges to the full-precision perplexity. The perplexity of `Bi-Mamba` on the wiki2 and C4 datasets is more stable than the perplexity on the PTB dataset. Notably, on the C4 dataset, the final perplexity of `Bi-Mamba` is even lower than the full-precision model, highlighting the superior performance of `Bi-Mamba`. Interestingly, early in training, our `Bi-Mamba` model surpasses GPTQ-3bit on the perplexity, demonstrating the effectiveness of binarization-aware training. Since the perplexity of GPTQ-2bit and BiLLM is extremely high on all datasets, we omit them in the figure and refer to Table 3 for detailed results of GPTQ-2bit and BiLLM. Moving to the downstream task evaluation, we first observe the catastrophic performance degradation of the binarized models, whose performance is

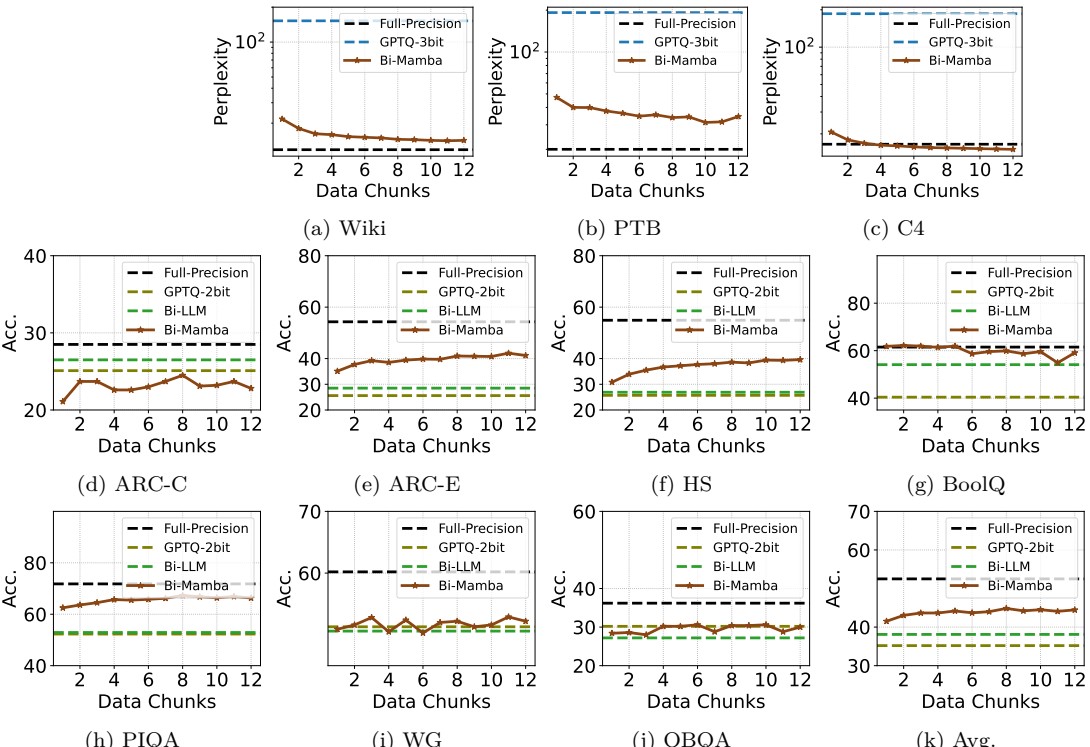

Figure 4: The training result dynamics of Downstream performance and perplexity of `Bi-Mamba`.

| Model | Zero-shot Acc. ↑ | | | | | | | | Perplexity ↓ | | |
|---|---|---|---|---|---|---|---|---|---|---|---|
| | BoolQ | PIQA | HS | WG | ARC-e | ARC-c | OBQA | Avg. | Wiki2 | PTB | C4 |
| Mamba-2-780M | 61.5 | 71.8 | 54.9 | 60.2 | 54.3 | 28.5 | 36.2 | 52.5 | 11.8 | 20.0 | 16.5 |
| Bi-Mamba (In_Proj) | 59.0 | 68.3 | 41.2 | 53.7 | 42.6 | 24.3 | 29.4 | 45.5 | 13.8 | 30.5 | 14.4 |
| Bi-Mamba (In_Proj + Out_Proj) | 58.5 | 68.0 | 41.6 | 52.0 | 42.4 | 24.3 | 30.6 | 45.3 | 13.4 | 32.4 | 14.5 |
| Mamba-2-1.3B | 64.3 | 73.7 | 59.9 | 61.0 | 60.4 | 33.1 | 37.8 | 55.8 | 10.4 | 17.7 | 14.8 |
| Bi-Mamba (In_Proj) | 62.1 | 71.7 | 50.4 | 53.8 | 49.5 | 26.8 | 33.0 | 49.6 | 10.7 | 26.0 | 12.0 |
| Bi-Mamba (In_Proj + Out_Proj) | 60.0 | 68.8 | 47.3 | 55.9 | 48.0 | 26.3 | 32.2 | 48.4 | 11.7 | 29.9 | 12.9 |
| Mamba-2-2.7B | 70.7 | 76.3 | 66.6 | 63.9 | 64.8 | 36.3 | 38.8 | 59.6 | 9.1 | 15.3 | 13.3 |
| Bi-Mamba (In_Proj) | 62.4 | 74.6 | 58.9 | 57.1 | 54.0 | 29.4 | 35.4 | 53.1 | 9.1 | 19.5 | 10.5 |
| Bi-Mamba (In_Proj + Out_Proj) | 58.0 | 72.5 | 54.3 | 56.1 | 51.4 | 29.1 | 32.6 | 50.6 | 10.0 | 21.9 | 11.3 |

Table 5: Performance comparison of partial-binarization and binarization of In_Proj and Out_Proj on perplexity and downstream tasks. The results show that the performance gap between partial-binarization and binarization of In_Proj and Out_Proj is not significant, indicating that the In_Proj and Out_Proj binarized model can maintain competitive performance with the partial binarization model.

even lower than the random results on many benchmarks, such as the results on ARC-E and ARC-C. This indicates that directly applying the naive binarization method destroys the ability of the full-precision model. However, after binarization-aware training, the model recovers the performance on all benchmarks and finally outperforms all baselines, including GPTQ-3bit, GPT-2bit, and Bi-LLM. This underscores the importance of binarization-aware training in achieving competitive results.

## 5.2 Binarization Space

In this section, we explore the binarization space of Mamba models and discuss the effect of binarizing each part. We conduct experiments that binarize the In_Proj and binarize both In_Proj and Out_Proj. The binarized models are trained with the same data. The results are shown in Table 5. Partial and full binarization are compared with the full-precision Mamba model on perplexity and downstream tasks. First, the results in the table show that partial binarization generally retains higher zero-shot accuracy compared

| Model | Tokens | GPU Hours | Zero-shot Acc. ↑ | | | | | | | | Perplexity ↓ | | |
|---|---|---|---|---|---|---|---|---|---|---|---|---|---|
| | | | BoolQ | PIQA | HS | WG | ARC-e | ARC-c | OBQA | Avg. | Wiki2 | PTB | C4 |
| Mamba-2 (130M, 16-bit) | 300B | 1180 | 55.1 | 64.0 | 35.3 | 52.6 | 47.4 | 24.1 | 30.6 | 44.2 | 20.0 | 35.1 | 25.2 |
| Mamba-2 (370M, 16-bit) | 300B | 2360 | 54.0 | 69.2 | 46.9 | 55.4 | 48.7 | 26.7 | 32.4 | 47.6 | 14.1 | 24.2 | 19.0 |
| Bi-Mamba (780M, 1-bit) | 105B | 4640 | 58.5 | 68.0 | 41.6 | 52.0 | 42.4 | 24.3 | 30.6 | 45.3 | 13.4 | 32.4 | 14.5 |
| Bi-Mamba (1.3B, 1-bit) | 105B | 5780 | 60.0 | 68.8 | 47.3 | 55.9 | 48.0 | 26.3 | 32.2 | 48.4 | 11.7 | 29.9 | 12.9 |
| Bi-Mamba (2.7B, 1-bit) | 105B | 7822 | 58.0 | 72.5 | 54.3 | 56.1 | 51.4 | 29.1 | 32.6 | 50.6 | 10.0 | 21.9 | 11.3 |

Table 6: Performance comparison of `Bi-Mamba` and full-precision pretrained Mamba-2 models in small sizes. All our `Bi-Mamba` models are better than Mamba-2-130M 16-bit model, which is equivalent to a 2.0B model in 1-bit. Moreover, `Bi-Mamba` 1.3B and 2.7B models achieve higher performance than Mamba-370M 16-bit model, which is equivalent to a 5.9B model in 1-bit, demonstrating the effectiveness of quantization-aware training instead of training a small model directly. The GPU hours are evaluated on A100 80G.

to full binarization. However, the performance gap between the partial and full binarized models is not significant. For instance, in the Mamba-2-2.7B model, partial binarization achieves an average accuracy of 53.1, while full binarization reduces this to 50.4. Across all model sizes, partial-binarized `Bi-Mamba` consistently outperforms full-binarized `Bi-Mamba` on most benchmarks, though shows minor performance degradation compared with full-precision models. It also suggests that fully binarization remains highly competitive and does not substantially lag. In terms of perplexity, the fully binarized model also performs comparably to the partial model. For example, in the Mamba-2-780M model, the C4 dataset perplexity for full-binarized `Bi-Mamba` (Fully) is 15.0, compared to 14.4 for partial-binarized `Bi-Mamba`, demonstrating that full binarization does not impose a significant perplexity increase. These findings highlight that the fully binarized model can maintain competitive performance with the partial binarization model, particularly in terms of perplexity, while still benefiting from greater storage and computational efficiency.

## 5.3 Comparison with Full Precision Small Models

Instead of binarization, one can train small models with full precision from scratch. We add the performance comparison of `Bi-Mamba` and small models pretrained with full precision, as shown in Table 6. We utilize the official pretrained weight from Mamba2, including models with 130M and 370M parameters pretrained with 300B tokens. 130M and 370M models in 16 bits are equivalent to 2.0B and 5.9B models in 1 bit, respectively. From the table, all `Bi-Mamba` models—including the 780M, 1.3B, and 2.7B variants trained on only 105B tokens—outperform the 130M full-precision models in average performance. Specifically, Mamba-2 130M obtains 44.2 accuracy on downstream tasks, with perplexities of 20.0, 35.1, and 25.2 on the Wiki, PTB, and C4 datasets, respectively. In comparison, the smallest `Bi-Mamba` model, with 780M parameters, achieves 45.3 accuracy on downstream tasks and perplexities of 13.4, 32.4, and 14.5 on Wiki, PTB, and C4, respectively. Moreover, `Bi-Mamba` 1.3B and 2.7B models achieve higher average performance than the full-precision 370M Mamba-2 model. Full-precision Mamba-2 370M gains 48.4 on average on downstream tasks. The `Bi-Mamba` 1.3B model outperforms the 370M Mamba-2 model, achieving 48.4 accuracy on downstream tasks. The results indicate that binarization with post-training is better than training a full-precision small model from scratch. Although `Bi-Mamba` includes extra training efforts due to the distillation compared with the small model, the overall GPU hours are comparable with the model with similar parameter size trained from scratch because of the lower training token requirement. Furthermore, our work is centered on the deployment phase, where binarized models can yield substantial benefits in terms of memory footprint, storage size, and inference efficiency—especially in CPU- or edge-constrained environments. As shown in Table 8, the fully binarized Bi-Mamba-2.7B model achieves 5× memory reduction, 3× faster throughput, and 3× lower energy consumption. Training-time cost is higher, but this is a one-time investment, amortized over repeated inference runs—particularly in latency-sensitive or resource-constrained deployment scenarios.

## 5.4 Storage Efficiency

`Bi-Mamba` is trained with full precision but can be saved as binary values in storage and inference. During training, we use $Sign(\cdot)$ function to obtain the binarized weights. Model binarization can significantly reduce the storage requirement on disk. Following Bi-LLM (Huang et al., 2024), we provide the theoreti-

| Model | Model Param. | Storage Size | Compress Ratio |
|---|---|---|---|
| Mamba-2 | 780M | 1.45GB | - |
| Bi-Mamba (InProj) | 780M | 0.63GB | 56.5% |
| Bi-Mamba (Full) | 780M | 0.22GB | 84.8% |
| Mamba-2 | 1.3B | 2.50GB | - |
| Bi-Mamba (InProj) | 1.3B | 1.01GB | 59.6% |
| Bi-Mamba (Full) | 1.3B | 0.33GB | 86.8% |
| Mamba-2 | 2.7B | 5.03GB | - |
| Bi-Mamba (InProj) | 2.7B | 2.01GB | 60.0% |
| Bi-Mamba (Full) | 2.7B | 0.55GB | 89.0% |

Table 7: Storage efficiency Bi-Mamba. Compared with partial binarization, full binarization can reduce the storage size significantly in all scales.

| Model | Model Param. | Memory | Tokens/s | Energy/128 Tokens |
|---|---|---|---|---|
| Mamba-2-16bit | 2.7B | 5.03GB | $24.59 \pm 2.26$ | 148.4J |
| Bi-Mamba | 2.7B | 1.04GB | $77.81 \pm 1.81$ | 44.9J |

Table 8: Resource consumption analysis of Bi-Mamba including memory, throughput and energy consumption on Mac M4 Pro CPU platform. Bi-Mamba reduces memory consumption significantly by more than $5\times$ and the energy consumption by more than 3 times. The throughput of inference is also increased to $3.16\times$, from 24.59 to 77.81.

cal storage requirement for our Bi-Mamba in different model sizes compared with full-precision models, as shown in Table 7. For each parameter size, the storage requirements for the original full-precision Mamba-2 model are substantially larger than those for the binarized Bi-Mamba, including partial and full binarization. Specifically, fully-binarized Bi-Mamba demonstrates the highest compression ratio, achieving reductions of more than 80%. In contrast, partial-binarized Bi-Mamba provides relatively moderate compression, ranging from 55% to 60%. This analysis highlights the efficiency of full binarization in significantly reducing storage requirements while maintaining the model parameter count, making it a highly storage-efficient alternative for large models.

### 5.5 Resource Consumption Analysis

We present the resource consumption analysis of Bi-Mamba, as shown in Table 8. We deploy our Bi-Mamba on the CPU of Mac M4 Pro Chip by implementing Bi-Mamba with llama.cpp framework[*]. The throughput is measured on text generation with 128 tokens generated. The computation is repeated and calculated over 5 runs with 8 CPU threads. For the energy computation, we utilize the asitop[*] tool to record the average CPU power consumption on a Mac system, as well as the overall computation clock time. The final energy is computed as the product of the average power consumption and the program's running time. The memory requirements for each model are evaluated with 2,048 tokens and a batch size of 128. Notably, Bi-Mamba reduces memory usage by more than $5\times$ and energy consumption by $3\times$, while increasing inference throughput to $3.16\times$—from 24.59 to 77.81. Moreover, energy consumption is reduced by more than $3\times$, from 148.4 J to 44.9 J for 128-token generation. These improvements in resource efficiency highlight Bi-Mamba's potential to significantly reduce computational costs in practice.

### 5.6 Results on Multilingual Benchmark

To show the performance on multilingual benchmark, we provide the results of Bi-Mamba on xwinograd in the table below to prove the generalization capability of our model, which is a multilingual benchmark including English, Chinese, Russian, Japanese, French and so on, as shown in Table 9. We evaluate all models with 2.7B parameters, except for OneBit, which has 7B parameters. Compared with both PTQ and

---

[*]https://github.com/ggml-org/llama.cpp.

[*]https://github.com/tlkh/asitop.

| Model | Param. | Xwinograd |
|---|---|---|
| Mamba-2-16bit | 2.7B | 75.2 |
| GPTQ-3bit | 2.7B | 64.8 |
| GPTQ-2bit | 2.7B | 49.2 |
| OneBit | 2.7B | 50.4 |
| Bi-Mamba | 2.7B | **68.0** |

Table 9: Performance comparison on Xwinograd, which is a multilingual benchmark.

| Model | Zero-shot Acc. ↑ | | | | | | | | Perplexity ↓ | | |
|---|---|---|---|---|---|---|---|---|---|---|---|
| | BoolQ | PIQA | HS | WG | ARC-e | ARC-c | OBQA | Avg. | Wiki2 | PTB | C4 |
| Mamba-2-780M | 61.5 | 71.8 | 54.9 | 60.2 | 54.3 | 28.5 | 36.2 | 52.5 | 11.8 | 20.0 | 16.5 |
| Bi-Mamba (No KD) | 50.5 | 65.8 | 37.8 | 50.9 | 39.7 | 23.8 | 30.4 | 42.7 | 14.9 | 30.9 | 15.6 |
| Bi-Mamba (KL-Div) | 56.8 | 66.5 | 38.1 | 51.6 | 39.8 | 22.7 | 28.2 | 43.3 | 15.0 | 27.3 | 15.6 |
| Bi-Mamba (Phi-3.5) | 49.6 | 66.8 | 39.0 | 53.2 | 40.6 | 23.7 | 30.8 | 43.4 | 14.5 | 27.3 | 16.6 |
| Bi-Mamba | 58.5 | 68.0 | 41.6 | 52.0 | 42.4 | 24.3 | 30.6 | 45.3 | 13.4 | 32.4 | 14.5 |

Table 10: Ablation study of Bi-Mamba. This table includes the performance comparison of different teachers and knowledge distillation strategies. Our autoregressive knowledge distillation brings improvement to the binarization-aware training regardless of the choice of teachers.

QAT methods, Bi-Mamba achieves the best performance on the multilingual benchmark, even surpassing the GPTQ-3bit model.

## 5.7 Ablation Study

In this section, we provide the ablation study of Bi-Mamba. As shown in Table 10, we conduct various ablation studies in the 780M model including the model without knowledge distillation (w/o KD in the table), the model utilizing the KL divergence loss as distillation loss (KL Div in the table) and the model using a different teacher model (Phi-3.5-instruct-mini (Abdin et al., 2024)). The results demonstrate the importance of each component, as removing knowledge distillation (KD) or using alternate loss functions (KL Div and Phi-3.5) significantly reduces performance compared to the full Bi-Mamba model, which achieves the best results across all metrics except PTB. Moreover, the stronger performance with Llama2-7B as a teacher compared with Phi-3.5 as a teacher indicates that the better the teacher is, the higher performance the student will achieve. Specifically, Bi-mamba trained with the original autoregressive loss obtains the lowest performance compared with other models trained with KD. The average accuracy on downstream tasks is 42.7, which is surpassed by the model trained with KL-Div loss, namely 43.3. With a different teacher, Phi-3.5, Bi-Mamba achieves similar performance as the model trained with Llama-2-7B as the teacher, demonstrating the effectiveness of our proposed autoregressive knowledge distillation.

## 6 Conclusion

We introduce Bi-Mamba, a scalable and efficient 1-bit Mamba architecture designed for large language models in multiple sizes: 780M, 1.3B, and 2.7B parameters. We begin by identifying the binarization space within the Mamba architecture. Then, Bi-Mamba models are trained from scratch on large datasets, similar to standard LLM pretraining, using an autoregressive distillation loss. Extensive language modeling experiments show that Bi-Mamba achieves competitive performance, only slightly lower than its full-precision counterparts (e.g., FP16 or BF16), while substantially reducing memory usage and computational cost compared to the original-precision Mamba. This study provided a novel, first accessible and low-bit framework with linear computational complexity, laying the foundation for developing specialized hardware optimized for efficient 1-bit Mamba-based LLMs.

## Limitations and Ethical Statements

While `Bi-Mamba` achieves competitive performance to full-precision models, there may still be trade-offs in accuracy, particularly in complex tasks that rely heavily on nuanced language understanding. Also, full deployment may require specialized hardware to maximize efficiency gains, limiting accessibility on standard hardware setups. On ethical part, reducing model precision could risk oversimplifying nuanced patterns in data, potentially amplifying biases present in the training data. Moreover, while `Bi-Mamba` reduces energy consumption during inference, training binary models from scratch can still be computationally intensive.

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

## Appendix

## A    More Experimental Results

### A.1    Full Results on Downstream Tasks

We present the performance dynamics of the 2.7B and 1.3B `Bi-Mamba` models during training in terms of perplexity and multiple downstream tasks, as shown in Figure 5 and 6. Each configuration demonstrates different trade-offs between model performance and computational efficiency of `Bi-Mamba` across various datasets.

We can observe that both the 2.7B and 1.3B `Bi-Mamba` models consistently outperform GPTQ-2bit and BiLLM after reaching a certain training stage. Specifically, `Bi-Mamba` generally exhibits a gradual decrease in perplexity, indicating the effectiveness of the training. In some cases such as `Bi-Mamba`-2.7B on C4 dataset, the perplexity is even better than in the full-precision Mamba models. In downstream tasks such as ARC-C, ARC-E, HS, and PIQA, `Bi-Mamba` consistently outperforms Bi-LLM, indicating that it is a more effective low-bit quantization approach for maintaining accuracy across various data sizes.

Additionally, the performance on average increases steadily with more training costs. The overall trend demonstrates that `Bi-Mamba` provides a robust alternative, balancing computational efficiency with competitive performance. This makes `Bi-Mamba` particularly valuable in resource-constrained environments where some trade-off in precision is acceptable. At the current training stage, the models have not yet fully converged, indicating potential for further performance gains.

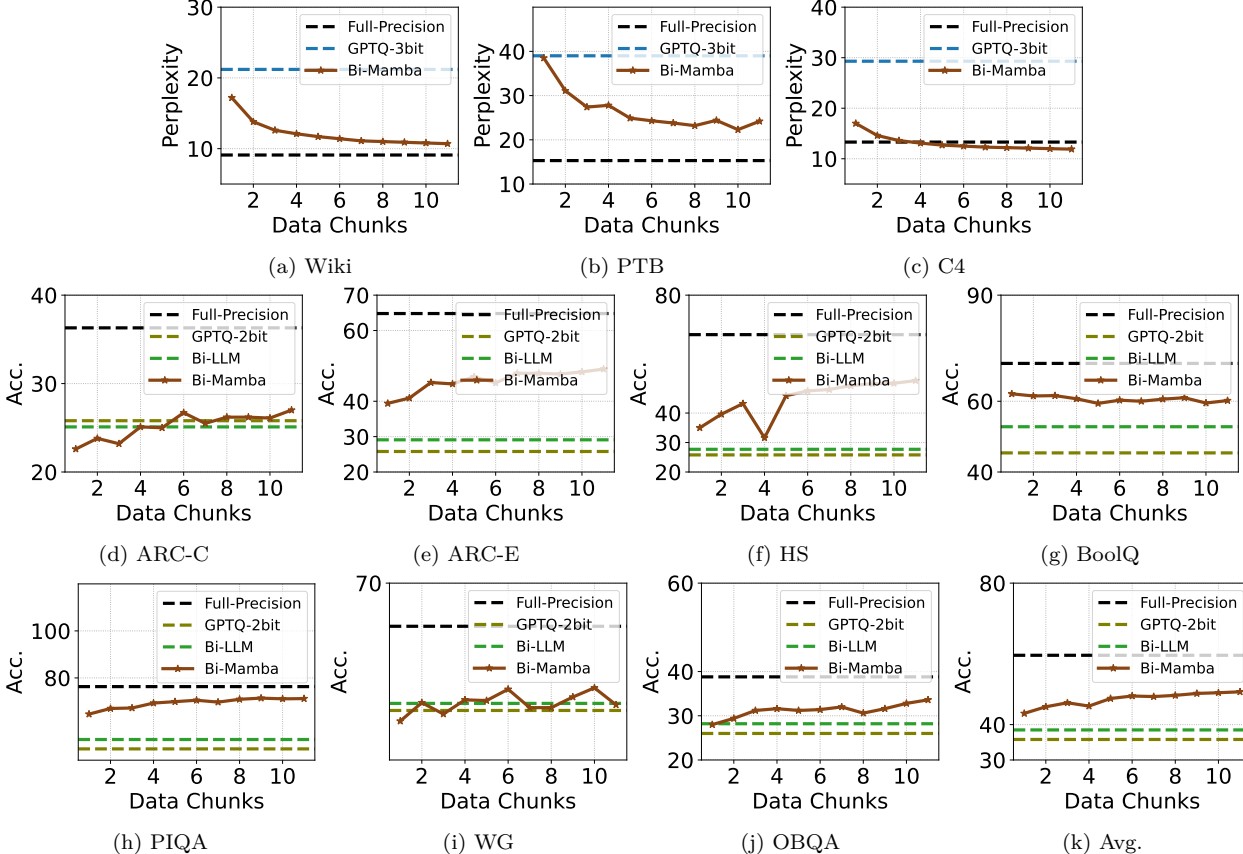

Figure 5: The downstream performance and perplexity curve of `Bi-Mamba`-2.7B with different training costs.

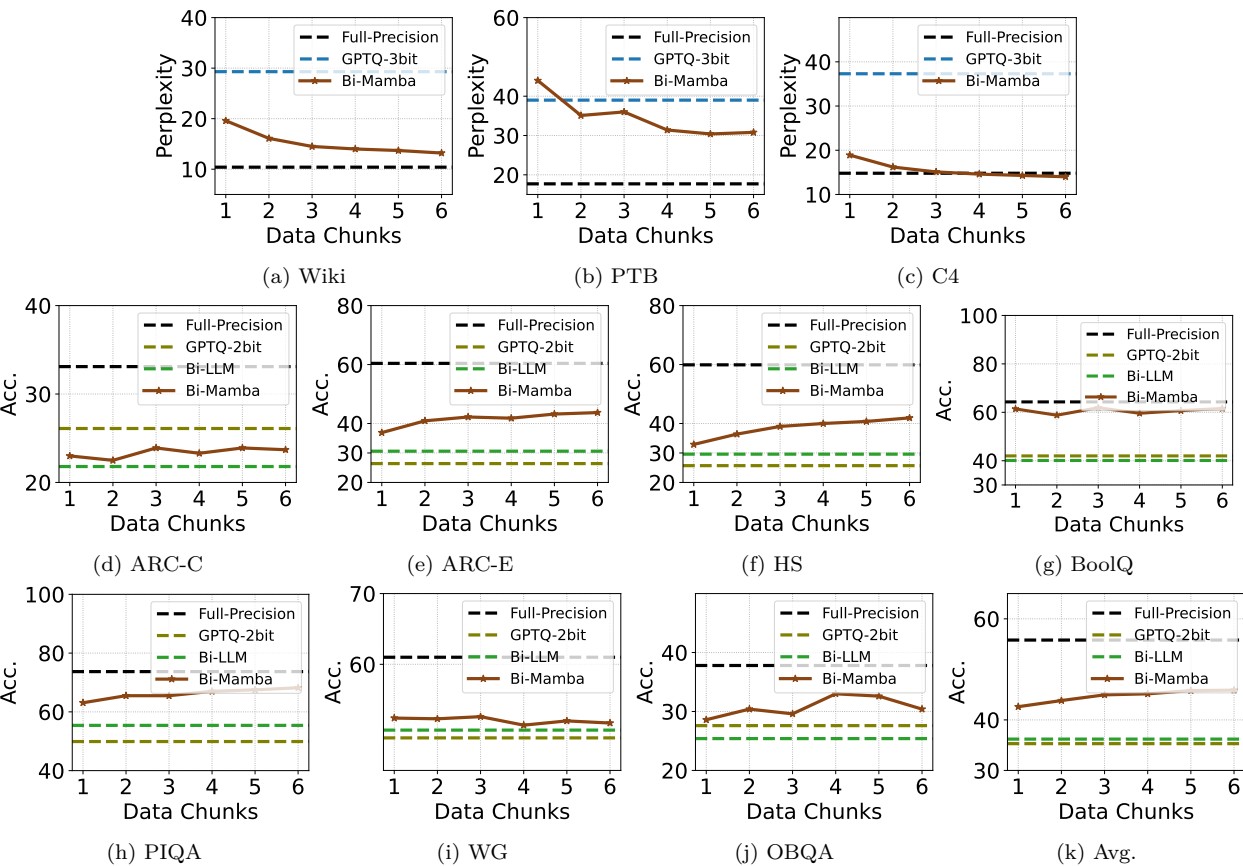

Figure 6: The downstream performance and perplexity curve of `Bi-Mamba`-1.3B with different training costs.

We provide the full results of the ablation study on downstream tasks, as shown in Table 10. Without knowledge distillation, the model obtains the lowest average zero-shot accuracy on downstream tasks. With knowledge distillation, `Bi-Mamba` achieves the highest accuracy on downstream tasks, demonstrating the effectiveness of our `Bi-Mamba`.

## A.2 Weight Distribution

We visualized the weight distributions of different modules in Mamba-2 (Orange histograms) and `Bi-Mamba` (Blue histograms), as shown in Figure 8, 9 and 10. We visualize the weight parameter distributions of different modules in the first (1st), mid (24th) and final (48th) layers of the corresponding 780M models. The input and output projection matrices are the values after re-scaling. Each pair of histograms compares how `Bi-Mamba` modifies the distribution of weights in different modules, no matter whether the module is binarized or not, illustrating the impact of `Bi-Mamba` on each module.

Specifically, in the first layer, the weight distribution of the original Mamba-2 such as *Conv1d.weight*, *Conv1d.bias* and *D* are tightly concentrated, indicating the strong focus on specific values. In contrast, the weight distribution in `Bi-Mamba` in the first layer is much divergent with additional peaks in the histograms such as in *A-log*, *Conv1d.bias* and *D*. The divergent weight distributions in `Bi-Mamba` suggest that `Bi-Mamba` intentionally captures broader values in the initial layers to retain sufficient information for binarized modules.

With more variability at the initial layers, `Bi-Mamba` can process the diverse initial features even in low-bit precision. In the mid-depth layers such as the 24th layer, the weight distribution of both the original Mamba and `Bi-Mamba` show similar patterns as in the first layer. However, the divergence is more moderate

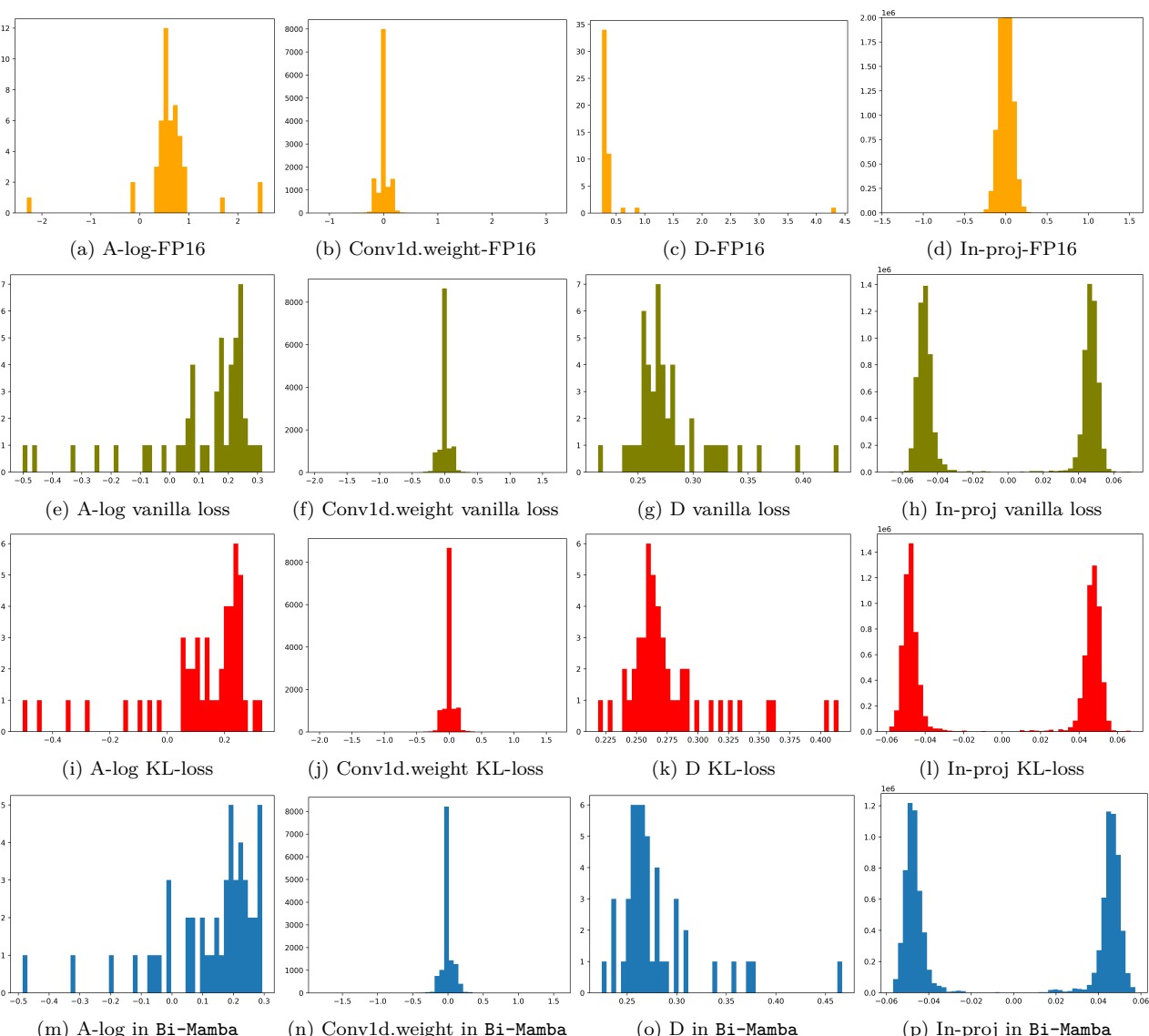

Figure 7: Distribution comparison of precision FP16 and different training objectives. It shows that different training objectives generate similar weight distributions.

compared with the divergence in the first layer. This suggests that in the intermediate layers, `Bi-Mamba` can refine the intermediate representation with generalization with binarized weights. Finally, in the last layer, the divergence patterns also remain while the distribution is much narrower compared with previous layers, reflecting a more concentrated range of values.

We also provide the weight distribution of different training objectives, as shown in Figure 7. With binarization, different training objectives including vanilla loss, KL-Divergence loss and our autoregressive loss, generate similar weight distribution after training.

The focused distribution helps to model to generate a stable and reliable final representation. In all, our `Bi-Mamba` includes a much wider distribution to capture more information at the beginning stages while the distribution tends to be more centralized progressively to output stable final results.

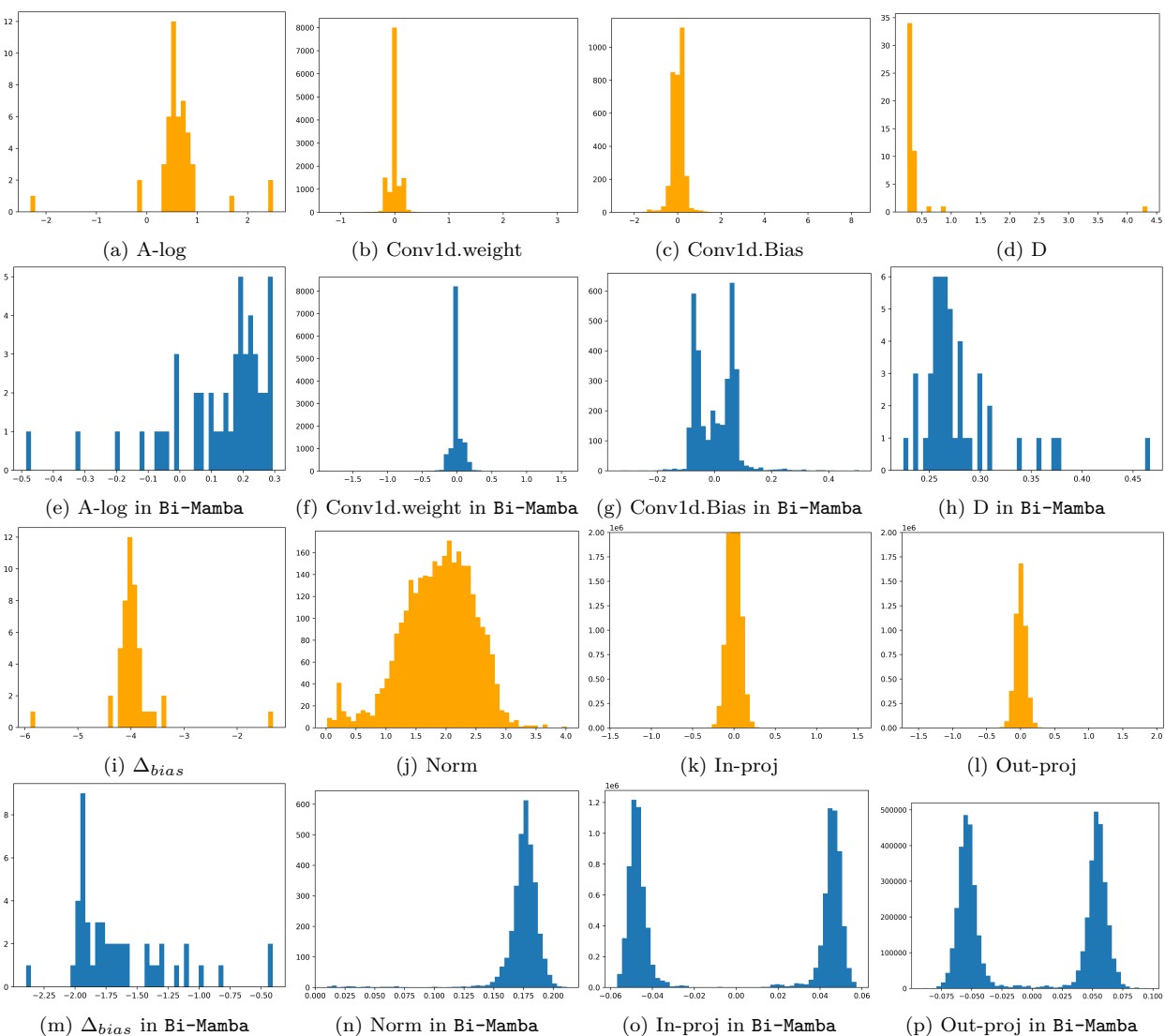

Figure 8: Distribution Comparison of each weight in Mamba-2 and `Bi-Mamba` modules at the 1st layer.

## B  Generation Case

We provide generation cases from our models in different scales including 2.7B, 1.3B and 780M, and other baseline models including GPTQ-3bit, GPTQ-2bit and Bi-LLM as shown in Figure 11, 12 and 13.

It is observed that Mamba-2 consistently produces coherent answers with meaningful semantic information but often repeats the content excessively. `Bi-Mamba`, while also retaining coherence and context after binarization, also shows repetition, especially in phrases. Nevertheless, `Bi-Mamba` is more robust than other baseline methods in preserving relevant information.

Post-training quantization methods, particularly at lower bit levels (e.g., GPTQ-2bit and 1bit BiLLM), tend to produce meaningless or garbled content. Specifically, GPTQ-3bit occasionally provides coherent starts but quickly devolves into repetitive or nonsensical text, indicating limited content understanding and generation ability after quantization. Other low-bit settings such as GPTQ-2bit and Bi-LLM generally fail to maintain coherence for generation, resulting in meaningless symbols generation, especially in smaller models.

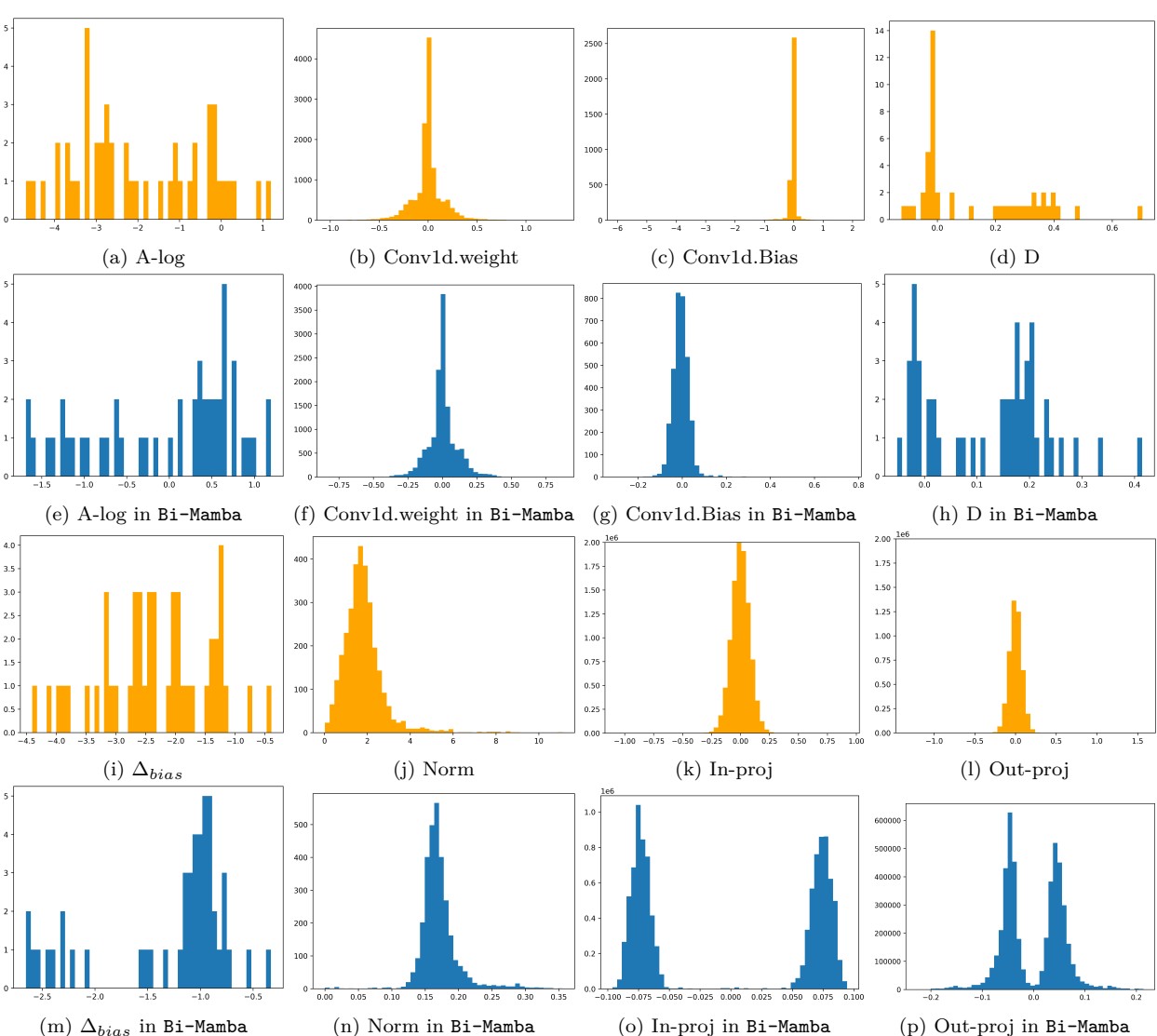

Figure 9: Distribution Comparison of each weight in Mamba-2 and `Bi-Mamba` modules at the 24th layer.

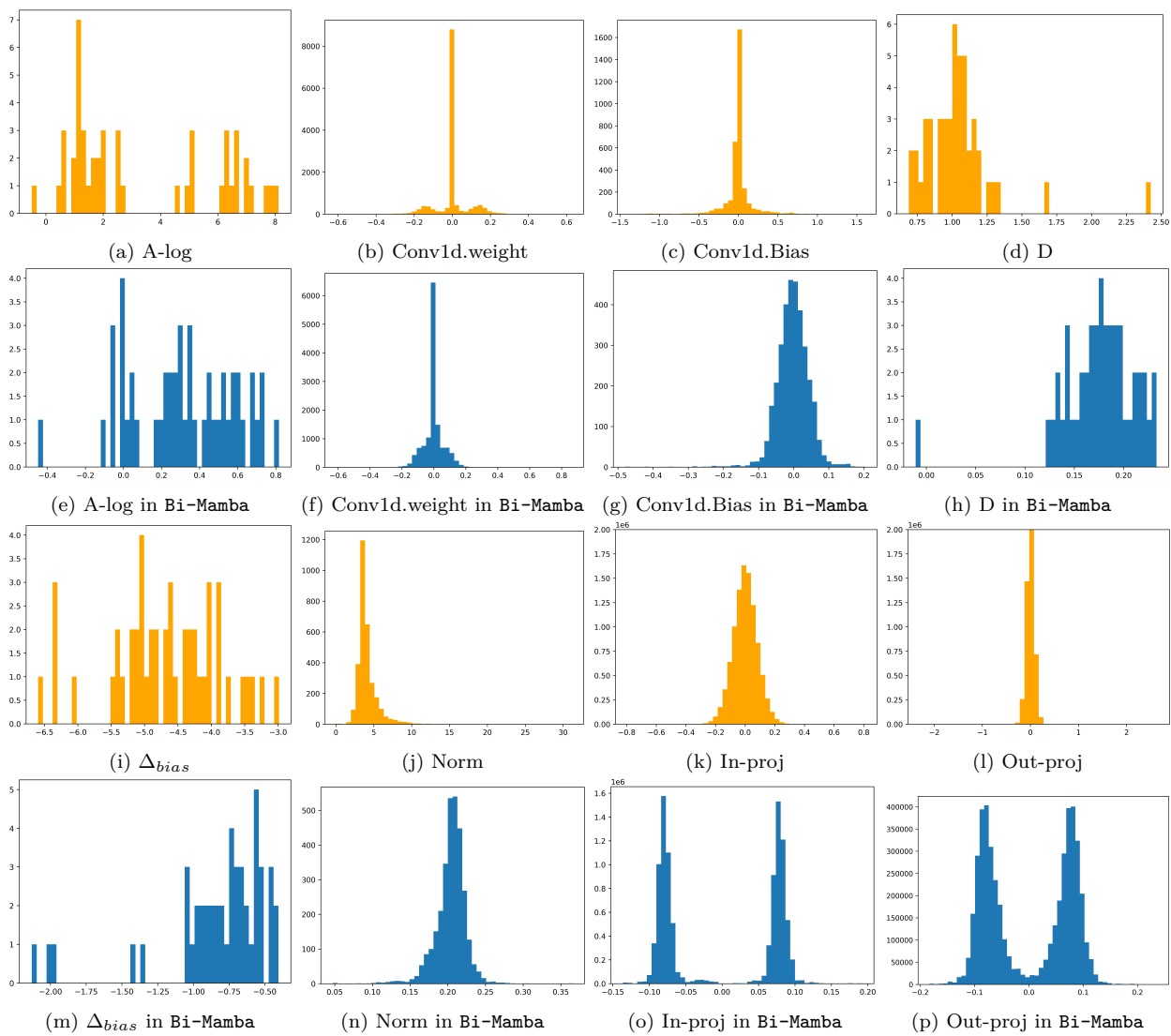

Figure 10: Distribution Comparison of each weight in Mamba-2 and `Bi-Mamba` modules at the 48th layer.

| | Prompt: Who is the world's most famous painter?\nAnswer: |
|---|---|
| **Mamba-2** | \nLeonardo da Vinci.\nLeonardo da Vinci was born in 1452 in the Italian city of Vinci.\nHe was a brilliant artist and scientist.\nHe was a great inventor and a great engineer.\nHe was a great musician and a great poet.\nHe was a great writer and a great philosopher.\nHe was a great mathematician and a great scientist.\nHe was a great architect and a great engineer.\nHe was a great painter and a great sculptor.\nHe was a great musician and a great poet.\nHe was a great writer and a great philosopher.\nHe was a great musician and a great poet. … **(REPEAT)** |
| **Bi-Mamba** (InProj) | 1. Leonardo da Vinci\n2. Leonardo da Vinci\n3. Leonardo da Vinci\n4. Leonardo da Vinci\n5. Leonardo da Vinci\n6. Leonardo da Vinci\n7. Leonardo da Vinci\n8. … **(REPEAT)** |
| **Bi-Mamba** (Fully) | 1. Leonardo da Vinci\n2. Leonardo da Vinci\n3. Leonardo da Vinci\n4. Leonardo da Vinci\n5. Leonardo da Vinci\n6. Leonardo da Vinci\n7. Leonardo da Vinci\n8. … **(REPEAT)** |
| **GPTQ** (3-bit) | The most famous painter in the world is Vincent Van Gogh, and the most famous artist in the world is Vincent Van Gogh.\nThe most famous artist in the world is Vincent Van Gogh, and the most famous artist in ktuska's world is Vincent Van Gogh.\nThe most famous artist in ktuska's world is ktuska's own painting, and the most famous artist in ktuska's world ktuska is ktuska's own painting.\nThe most famous artist in ktuska ktuska's world is ktuska's own painting, ktuska's own painting is ktuska's own painting, … **(REPEAT)** |
| **GPTQ** (2-bit) | /bb,fitting C- (umC of price in - pil dst\n, WHE,\n\n\naccordMeierin ( - andY...]({ORG.a\n*F000001T (\n  and-wx20 or,idasa \n r |
| **BiLLM** | \n* *\ufffdER is answors_\n\n\n%%%%\n\n( )\n%%%%\n\n( )%%%%\ufffdER\n\n%%%% … **(MEANINGLESS)** |

Figure 11: The generation cases for 2.7B models.

| Prompt: How did Julius Caesar die?\nAnswer: | |
|---|---|
| Mamba-2 | \nHe was stabbed by a Roman soldier named Sextus Pompey.\n\nA:\n\nHe was stabbed by a Roman soldier named Sextus Pompey.\n\nThe answer is:\n\nHe was stabbed by a Roman soldier named Sextus Pompey.\n\nThe word "stabbing" is a verb, and the verb "stabbing" is a noun.\n\nA:\n\nHe was stabbed by a Roman soldier named Sextus Pompey.\n\nThe word "stabbing" is a verb, and the verb "stabbing" is a noun.\n\nA:\n\nHe was stabbed by a Roman soldier named Sextus Pompey.\n\n |
| Bi-Mamba (InProj) | 1. He was killed by a Roman soldier.\n2. He was killed by a Roman soldier.\n3. He was killed by a Roman soldier.\n4. He was killed by a Roman soldier.\n5. … **(REPEAT)** |
| Bi-Mamba (Fully) | 100 years ago\nAnswer: 100 years ago\nAnswer: 100 years ago\nAnswer: 100 years ago\nAnswer: 100 years ago\nAnswer: 100 years ago\nAnswer: 100 years ago\nAnswer: 100 years ago … **(REPEAT)** |
| GPTQ (3-bit) | \nHow did Julius Caesar die?: \nHow did Julius Caesar die?: \nHow did Julius Caesar die?: \n\n \nHow did Julius Caesar die?: \nHow did Julius Caesar die?: \n\n \nHow did Julius Caesar die?: \nHow did Julius Caesar die?: \n\n … **(REPEAT)** |
| GPTQ (2-bit) | LinafterinfalleverP .'\n HCl N-\n,[ACex\ns\ufffd (nothingyil or\nnil_box\nPdis+one ocks .\nnumbers\n a <T data\u00a0Problem PermianMetricsorvdots:quoLQCb … **(MEANINGLESS)** |
| BiLLM | \n\n"Juluccan"," a) \n\n"auc\n\n"a\n\n"a … **(MEANINGLESS)** |

Figure 12: The generation cases for 1.3B models.

| Prompt: Give three tips for staying healthy.\nAnswer: | |
|---|---|
| Mamba-2 | \n1. Eat a healthy diet.\n2. Exercise regularly.\n3. Get enough sleep.\n\nAnswer: \n1. Eat a healthy diet.\n2. Exercise regularly.\n3. Get enough sleep.\n\nAnswer: \n1. Eat a healthy diet.\n2. Exercise regularly.\n3. Get enough sleep.\n\n … **(REPEAT)** |
| Bi-Mamba (InProj) | 1. Get enough sleep.\n2. Get enough exercise.\n3. Get enough rest.\nAnswer: 1. Get enough sleep. 2. Get enough exercise. 3. Get enough rest.\nAnswer: 1. Get enough sleep. 2. Get enough rest. 3. Get enough exercise.\n … **(REPEAT)** |
| Bi-Mamba (Fully) | 1. Eat a healthy diet.\n2. Eat a healthy diet.\n3. Eat a healthy diet.\n4. Eat a healthy diet.\n5. Eat a healthy diet.\n6. Eat a healthy diet.\n7. Eat a healthy diet.\n8. … **(REPEAT)** |
| GPTQ (3-bit) | \n1. I was a vegetarian for a while, so I a little bit of a fattie, so I a little a a a a a a a a a a a a a a a a a a a a a a a a a a aa a a a a a a a a a a a aa a a a … **(MEANINGLESS)** |
| GPTQ (2-bit) | .all balcon Vphabetf]5,-, tos- conflicbane-n*~(,-V--, and\nly l and U alleruckland e/ and. at E'ipel _/rains Gesch- a both/ for\n-aceutics. arrangatase … **(MEANINGLESS)** |
| BiLLM | \n.\n.\n;,\n]:\n:.\n.]\n:.\n.\n.\n:.\n.]\n:.\n.}\n]:\n:.\n.} … **(MEANINGLESS)** |

Figure 13: The generation cases for 780M models.

