# OpenReview forum: "Bi-Mamba: Towards Accurate 1-Bit State Space Model"
_TMLR — Accepted by TMLR_

### Review · Reviewer_VSpb · 2025-06-18

**Summary Of Contributions:**

This paper studies how to compress parameter variables to 1 bit within the State-Storage Models, specifically adapting the Mamba architecture.  The paper demonstrates improved performance over other 1 or 2/4 bit compression schemes for comparable models, but a drop in performance compared to the non-compressed models.

The Mamba architecture this paper studies is from a couple years ago, and while much of the work is un-published, it has 1000s of citations.  It is based on a Selective State Space Model that is a potential replacement for transformers that has linear (as opposed to quadratic) dependence on the context window.
This paper shows that the architectures are also more amenable to quantization than other models.

**Audience:**

Yes

**Claims And Evidence:**

Yes

**Requested Changes:**

Please explain Table 4 better, improve or remove un-needed visualizations, and rethink the work "exponentially"

**Strengths And Weaknesses:**

Strengths:
 + extensive experimental section on various parameter sizes (760M, 1.3B, 2.7B), many performance data sets, several quantized comparisons, including quantized GPT models.

 + clearly better accuracy (especially as factor of space saving) compared to other models.

 + nice discussion and empirical results on space and energy savings.

 + Seems to be important step towards effective highly quantized models.  If this is a direction the field will go to (for space, energy savings) then this paper will be important.

 + Does a nice job trying to explain the Mamba architecture to make the paper more self-contained.


Weaknesses:
 - Although, this appears the best performing of the quantized methods, it appears to consistently do worse (and significantly worse) than the non-quantized models.

 - Table 4 seems interesting, since Bi-Mamba sometimes outperforms Mamba-2-16bit (does this means it improves upon the non-quantized model!?!).  But while referenced twice in the text, this Table is not clearly explained.

 - Some of the visualizations felt pointless.  In particular the lifted pie charts in Figure 2, as well as the distribution plots in that figure.  And the radio plots in Figure 3.



Notes:
 - The second paragraph (page 1) uses the term "exponentially" in a way I think does not describe an exponential phenomenon-- maybe polynomial?

---

> ### Author Response · Authors · 2025-07-23
> **Response to Reviewer VSpb**
>
> Thank you for your detailed and constructive review. All the suggested comments have been included and revised in the updated version of the submission. We further reply to your concerns in the following:
>
> > W1:  Although, this appears the best performing of the quantized methods, it appears to consistently do worse (and significantly worse) than the non-quantized models.
>
> This is expected, improving inference efficiency inevitably comes with some performance trade-off. However, the performance drop here is very small and well worth it. This is the case for almost all quantization methods. Since binarization is an extreme form of quantization, such a performance loss is even more understandable.
> With the same number of parameters, a binary model naturally has less representational capacity compared to its full-precision counterpart. Therefore, it is expected that the binary model may underperform the full-precision version. However, as shown in our Table 6, when compared to smaller full-precision models with a similar parameter budget, our Bi-Mamba consistently outperforms them.
>
> > W2: Table 4 seems interesting, since Bi-Mamba sometimes outperforms Mamba-2-16bit (does this means it improves upon the non-quantized model!?!). But while referenced twice in the text, this Table is not clearly explained.
>
> Thanks for the comments. Yes, Bi-Mamba outperforms the non-quantized model on HumanEval and GSM8K. We believe the reason is the higher quality of pretraining data. Bi-Mamba applied Amber dataset, which is a high-quality dataset compared with Pile dataset used in the original Mamba model. Moreover, after instruction-tuning on OpenOrca dataset, the performance is further improved, demonstrating that Bi-Mamba can serve as a strong base model. We have added more clarification in the revision.
>
> > W3: Some of the visualizations felt pointless. In particular the lifted pie charts in Figure 2, as well as the distribution plots in that figure. And the radio plots in Figure 3.
>
> The lifted pie charts in Figure 2: As Mamba architecture is quite different from Transformer and this is the first work to explore its binarization potential, we would like to understand the binarization space of Mamba model and provide the pie charts to show the parameter allocation clearly as the binary candidate spaces.
>
> The distribution plots in Figure 2: We aim to investigate the difference in parameter distributions between post-binarization and binarization-aware training on Mamba architecture. This will help demonstrate that binarization-aware training may be a more feasible solution for Binary Mamba.
>
> The radio plots in Figure 3: We provide the radio plots to offer an intuitive comparison with baseline methods across different scales.
> We have revised them carefully in our new version, please check them out. Thanks.
>
>
> > Note: The second paragraph (page 1) uses the term "exponentially" in a way I think does not describe an exponential phenomenon-- maybe polynomial?
>
> Thanks for pointing this out. Yes, it should be polynomial. We have revised it accordingly in the revision.
>
> > R1: Please explain Table 4 better, improve or remove un-needed visualizations, and rethink the work "exponentially"
>
> Thanks for the suggestion. We have revised our paper carefully following the reviewer’s comments. Please check them out in our newest uploaded version.

---

### Review · Reviewer_hjFX · 2025-06-26

**Summary Of Contributions:**

The authors convincingly show that, at least for simple multiple-choice data sets, quantization of Mamba-class LLMs should be carried out with quantization-aware training (QAT) rather than post-training quantization (PTQ).  In particular, they show that a recently introduced QAT binarization scheme called FBI-LLM yields better zero-shot accuracies and perplexities than those achieved under various PTQ schemes.  The authors also support their argument in favor of this scheme with several ancillary analyses, showing (e.g.) that comparably sized (on disk), 16-bit, Mamba models perform worse than binarized Mambas.  The model-compression ratios are on the order of 85--90%, and the binarization scheme allows (in theory) for the replacement of many multiplications with binary operations, so the results have important implications for low-energy, storage-efficient LLMs.

**Audience:**

Yes

**Claims And Evidence:**

No

**Requested Changes:**

The authors should (critical)

- clarify what we learn from the success of FBI-LLM applied to Mamba models, including comparing it to other quantization-aware schemes applied to Mamba
- make an apples-to-apples comparison for BitNet (or as close as possible, since BitNet uses 1.58 bits rather than just 1)
- report the perplexities for OneBit

Ideally, they would also (strengthen)
- use statistical tests when comparing model performance
- address my other minor concerns

**Strengths And Weaknesses:**

I have described the strengths of the paper under the SUMMARY; here I list weaknesses.


MAJOR: What exactly have we learned from this study?  The field as a whole has moved toward QAT and away from PQT (whenever the cost of training is not a factor); this study simply applies that lesson to Mamba.

Perhaps the punchline is supposed to be the precise technique used for binarization, FBI-LLM.  But there are obstacles to this.  First, the scheme is not new to this paper (it was introduced in another paper last year, but applied to transformers).  Second, although the technique differs on some points from other QAT schemes (e.g., FBI-LLM learns real-valued scaling and shift parameters *severally* for each column of the matrix), we don't get much intuition about which aspects of the technique are important.  In particular, no comparison is made to any other QAT technique applied to Mamba models.

The authors also argue that their Bi-Mamba outperforms quantized transformer models, which would certainly be interesting.  But the results reported on this question have problems:

First of all, FBI-LLM is evidently (slightly) more effective when applied to Mamba than to when applied to transformers, it's original application.  This is potentially interesting---is there an explanation?

Second, in comparing to BitNet, the authors use the reported results from that paper [Ma et alia 2024].  But that version of BitNet quantizes the activations to 8 bits (i.e., in addition to quantizing the weights), in contrast to the full-precision activations in Bi-Mamba; and was trained only on 100B tokens of the RedPajama dataset, in contrast to the 1.2T tokens used to train the Bi-Mamba models.  As it stands, the BitNet models have very similar accuracies to Bi-Mamba, so it seems quite plausible that on equal footing, BitNet would outperform Bi-Mamba.  Furthermore, perplexities are not reported in this MS for BitNet, ostensibly because they were not reported in the original paper---but they are, in Table 1, and indeed are lower than those achieved by Bi-Mamba.

The other point of comparison for QAT transformers is OneBit, but perplexity isn't reported for it.  Why not?  (The authors indicate that they have access to the weights from that model.)


MINOR:
I realize that this area of the literature does not much make use of statistical tests, but it's never too late to start.  Surely there is some variation in performance across multiple re-trainings (or even just on different subsets of the test data), and ideally the authors would produce *distributions* of accuracies (and perplexities), and then compare the distributions for different training schemes with a t-test or the like.

Can the authors confirm that when applying the PQT schemes (GPTQ, SqueezeLLM, etc.) to Mamba, they only quantized the input and output projections, as in Bi-Mamba?

What is the bold font in Table 3 supposed to indicate?  Bi-Mamba's zero-shot accuracy is bolded even when it is not the highest.

Rows in the table that are copied from published papers (as for BitNet, e.g.), as opposed to reproduced by the authors, should be marked as such in the table.

Section 5.2: I'm not sure we learn much from this section.  Table 5 mostly just repeats the data from Table 3.  The only addition is a Bi-Mamba in which only the in-projection has been binarized.  Is the point simply that binarizing even less, namely not binarizing the output projection, doesn't really improve performance?  This is a somewhat arbitrary datum.  (More useful might be to see a plot of, say, performance vs. compression rato---not just for these two point, but also for binarizing *more* of the network.  Then we would presumably see clearly that the authors have chosen the right fraction of the network to binarize.)  It's also confusing that the Bi-Mamba model in which (only) the input and output projections are binarized is here called "fully binarized."

Section 5.3: The Bi-Mamba models achieve slightly higher accuracies than the comparable small Mambas, but require far more training time.  Is the energy trade-off worthwhile here?


Clarity: Some text is sometimes repetitive (the quadratic computation cost of transformer is repeated several times) and could be much reduced, and there are some confusing sentences (I note some below).  But the bigger issue is the grammatical errors, which can be found in roughly half of the sentences.  I think it would be unfortunate if this manuscript (whether it is published in TMLR or elsewhere) didn't receive proper attention because it is hard to read.  I strongly encourage the authors to have the MS read and edited by a native speaker of English---or, if that is impossible, at least an LLM.

"...on data volume as regular LLM pertaining using an autoregressive distillation loss"
I don't understand this clause.

"Prior works...."  In this context, "work" is a mass noun and shouldn't be pluralized.

"tenary weights" -> "ternary weights"?

"h_t then represents the derivative of the hidden state..."
Why write it as h_t?  If Eq. 1 is supposed to describe the continuous-time dynamics, it should just be written $$\dot h_t = Ah_t + B x_t$$ (etc.).

"Since B and C are derived from the linear projection of each layer’s inputs, and the language model head is tied to the embedding layer, these modules are ignored in our design."
I don't understand this sentence.

etc.

---

> ### Author Response · Authors · 2025-07-23
> **Response to Reviewer hjFX [1/4]**
>
> Thank you for your detailed and constructive feedback. All the suggested comments have been included and revised in our updated paper. We further reply to each of your questions in the following:
>
> >W1: MAJOR: What exactly have we learned from this study? The field as a whole has moved toward QAT and away from PQT (whenever the cost of training is not a factor); this study simply applies that lesson to Mamba.
>
> Mamba’s architecture and mechanisms differ significantly from conventional Transformers. Before our work, there had been no systematic study on which parameters in Mamba can be binarized and which components can tolerate binarization with minimal performance loss. We believe this is a key contribution of our work, something others can learn from this study. Our work is not simply to apply existing binarization lessons to Mamba, but to carefully consider and address the unique characteristics of Mamba itself.
> Specifically, our approach goes beyond standard QAT. We explicitly identify the binarization space within Mamba, selectively binarizing only the linear input/output projections while preserving the embeddings to maintain semantic representation. Additionally, we introduce an autoregressive distillation scheme adapted to Mamba’s recurrent state-space design. These design decisions are non-trivial, as the signal propagation in state-space models differs fundamentally from that in Transformers.
>
> >W2: Perhaps the punchline is supposed to be the precise technique used for binarization, FBI-LLM. But there are obstacles to this. First, the scheme is not new to this paper (it was introduced in another paper last year, but applied to transformers). Second, although the technique differs on some points from other QAT schemes (e.g., FBI-LLM learns real-valued scaling and shift parameters severally for each column of the matrix), we don't get much intuition about which aspects of the technique are important. In particular, no comparison is made to any other QAT technique applied to Mamba models.
>
> We appreciate the reviewer’s careful reading. Our contribution lies in successfully exploring binarization for the Mamba architecture and showing that binarization-aware training can scale to 2.7B-parameter state-space models, which operate under fundamentally different dynamics than Transformers. To further address the reviewer’s concern, we now include a direct comparison with OneBit. We follow the same QAT approach as OneBit for the Mamba architecture, using identical hyperparameters and the same training data:
>
> | Methods           | Wiki | PTB  | C4   | PIQA | HS   | Wino | Arc_c | OBQA | GSM8k | Humaneval |
> |-------------------|:------:|:------:|:------:|:------:|:------:|:------:|:-------:|:------:|:-------:|:-----------:|
> | 780M-OneBit-Mamba | 26.3 | 39.6 | 27.4 | 63.3 | 38.6 | 54.4 | 24.4  | 28.6 | 0.1  | 0.0      |
> | 780M-Bi-Mamba     | 13.4 | 32.4 | 14.5 | 68.0 | 41.6 | 52.0 | 24.3  | 30.6 | 1.3  | 0.9      |
> | 1.3b-OneBit-Mamba | 20.9 | 30.9 | 23.4 | 65.3 | 43.1 | 55.1 | 25.0  | 30.6 | 0.4  | 0.0       |
> | 1.3b-Bi-Mamba     | 11.7 | 29.9 | 12.9 | 68.8 | 47.3 | 55.9 | 26.3  | 32.2 | 2.0  | 0.7      |
> | 2.7b-OneBit-Mamba | 16.5 | 24.7 | 19.8 | 68.9 | 50.3 | 60.5 | 30.0  | 33.0 | 0.9  | 0.0      |
> | 2.7b-Bi-Mamba     | 10.0 | 21.9 | 11.3 | 72.5 | 54.3 | 56.1 | 29.1  | 32.6 | 1.3  | 0.9      |
>
> We show the Bi-Mamba trained models with similar tokens in the table above for comparison. We can observe that with the same training settings on Mamba architecture, Bi-Mamba outperforms OneBit on both perplexity and downstream tasks.

---

> ### Author Response · Authors · 2025-07-23
> **Response to Reviewer hjFX [2/4]**
>
> >W3: The authors also argue that their Bi-Mamba outperforms quantized transformer models, which would certainly be interesting. But the results reported on this question have problems: First of all, FBI-LLM is evidently (slightly) more effective when applied to Mamba than when applied to transformers, its original application. This is potentially interesting---is there an explanation? Second, in comparing to BitNet, the authors use the reported results from that paper [Ma et alia 2024]. But that version of BitNet quantizes the activations to 8 bits (i.e., in addition to quantizing the weights), in contrast to the full-precision activations in Bi-Mamba; and was trained only on 100B tokens of the RedPajama dataset, in contrast to the 1.2T tokens used to train the Bi-Mamba models. As it stands, the BitNet models have very similar accuracies to Bi-Mamba, so it seems quite plausible that on equal footing, BitNet would outperform Bi-Mamba. Furthermore, perplexities are not reported in this MS for BitNet, ostensibly because they were not reported in the original paper---but they are, in Table 1, and indeed are lower than those achieved by Bi-Mamba.
>
> Thanks for raising these concerns, we clarify: 1) We conjecture the improved performance on Mamba setting may stem from the architectural differences between state-space models and attention-based Transformers. Mamba models operate without self-attention and do not rely on key-query-value (KQV) dynamics. This makes them more robust to quantization and binarization of linear layers, especially when combined with binarization-aware training.
> 2) For the training data, our current compared models are also trained using ~100B from RedPajama, the same size as BitNet, as we mentioned in our Appendix. 3) BitNet only reports the average perplexity of C4 and Wikitext2 without the separate results and results on PTB. The average perplexity comparison is:
> | **Method**    | **Parameter Number** | **Perplexity** |
> |---------------|:----------------------:|:----------------:|
> | BitNet-1.58b  | 3B                   | 9.91           |
> | Bi-Mamba      | 2.7B                 | 10.65          |
>
> Bi-Mamba achieves perplexity comparable to that of BitNet. BitNet-1.58b shows slightly better perplexity, which we attribute to its larger size—3B parameters compared to Bi-Mamba's 2.7B—and its higher precision at 1.58 bits.
> >W4: The other point of comparison for QAT transformers is OneBit, but perplexity isn't reported for it. Why not? (The authors indicate that they have access to the weights from that model.)
>
> Thank you for pointing this out. We provide the perplexity of OneBit:
> | **Methods** | **Parameters** | **Wiki2** | **PTB** | **C4** |
> |-------------|:-----------:|:----------:|:---------:|:--------:|
> | OneBit      | 6.7B           |      10.2 | 18.2    | 11.6   |
> | Bi-Mamba    | 2.7B       |      10.0 | 21.9    | 11.3   |
>
> Bi-Mamba achieves lower perplexity on both Wiki2 and C4 while OneBit obtains better perplexity on PTB. We suspect the reason is the differences in training data and the fact that OneBit uses more parameters.
>
> >W5: I realize that this area of the literature does not much make use of statistical tests, but it's never too late to start. Surely there is some variation in performance across multiple re-trainings (or even just on different subsets of the test data), and ideally the authors would produce distributions of accuracies (and perplexities), and then compare the distributions for different training schemes with a t-test or the like.
>
> Thanks for the kind comments. Following the reviewer’s suggestion, we provide the variation on different subsets of test data 10 times in the table below, as the LLM involves large-scale pretraining, it is costly to perform multiple retrainings on full data.
>
> | Size | BoolQ        | PIQA        | HS           | WG          | ARC-e       | ARC-c       | OBQA        | Avg           | Wiki2         | PTB           | C4            |
> |------|--------------|-------------|--------------|--------------|--------------|--------------|-------------|----------------|----------------|----------------|----------------|
> | 780M | 58.5±0.039   | 68.0±0.471  | 41.6±0.077  | 52.0±0.035   | 42.4±0.084  | 24.3±0.106  | 30.6±0.459  | 45.3±0.072     | 13.4±0.124     | 32.4±0.095    | 14.5±0.116     |
> | 1.3B | 60.0±0.025   | 68.0±0.435  | 47.3±0.021   | 55.9±0.029  | 48.0±0.090 | 26.3±0.102  | 32.2±0.474  | 48.4±0.085     | 11.7±0.111     | 29.9±0.116     | 12.9±0.105     |
> | 2.7B | 58.0±0.011  | 72.5±0.448  | 54.3±0.035  | 56.1±0.068   | 51.4±0.104  | 29.1±0.112  | 32.6±0.493  | 50.6±0.104     | 10.0±0.105     | 21.9±0.114     | 11.3±0.121     |
>
> We have also updated this in the revised paper.
> >W6: Can the authors confirm that when applying the PQT schemes (GPTQ, SqueezeLLM, etc.) to Mamba, they only quantized the input and output projections, as in Bi-Mamba?
>
> Yes, these methods follow Bi-Mamba’s scheme and only quantize the input and output projections.

---

> > ### Author Response · Authors · 2025-07-23
> > **Response to Reviewer hjFX [3/4]**
> >
> > >W7: What is the bold font in Table 3 supposed to indicate? Bi-Mamba's zero-shot accuracy is bolded even when it is not the highest.
> >
> > Thanks for pointing this out. Bi-Mamba is not the highest since we added the 3-bit results of other methods, including GPTQ and SqueezeLLM. We have made some changes in this table to make the comparison clearer.
> >
> > >W8: Rows in the table that are copied from published papers (as for BitNet, e.g.), as opposed to reproduced by the authors, should be marked as such in the table.
> >
> > We have revised this to clearly indicate which results are reproduced by us and which are directly copied from the original papers. We also provided additional implementation details and comparisons to make the statements clearer. Please refer to Table 3 in our revision.
> > > Section 5.2: I'm not sure we learn much from this section. Table 5 mostly just repeats the data from Table 3. The only addition is a Bi-Mamba in which only the in-projection has been binarized. Is the point simply that binarizing even less, namely not binarizing the output projection, doesn't really improve performance? This is a somewhat arbitrary datum. (More useful might be to see a plot of, say, performance vs. compression ratio---not just for these two point, but also for binarizing more of the network. Then we would presumably see clearly that the authors have chosen the right fraction of the network to binarize.) It's also confusing that the Bi-Mamba model in which (only) the input and output projections are binarized is here called "fully binarized."
> >
> > Thanks for your careful reviews. Yes, these results show that binarization input and output projection together can yield similar performance as the model only binarizes the input projection. Since the input and output projection matrices account for the vast majority of model parameters (over 90%, as shown in Table 1), our experiments focus on binarizing these two components. Finally, we revise the paper accordingly to make our statement clearer.
> >
> > >Section 5.3: The Bi-Mamba models achieve slightly higher accuracies than the comparable small Mambas, but require far more training time. Is the energy trade-off worthwhile here?
> >
> > Yes, as shown in Table 8, the fully binarized Bi-Mamba-2.7B model achieves 5$\times$ memory reduction, 3$\times$ faster throughput, and 3$\times$ lower energy consumption. Training-time cost is relatively high, but this is a one-time investment, amortized over repeated inference use, especially in latency-sensitive or resource-limited deployment scenarios
> >
> > > Clarity: Some text is sometimes repetitive (the quadratic computation cost of transformer is repeated several times) and could be much reduced, and there are some confusing sentences (I note some below). But the bigger issue is the grammatical errors, which can be found in roughly half of the sentences. I think it would be unfortunate if this manuscript (whether it is published in TMLR or elsewhere) didn't receive proper attention because it is hard to read. I strongly encourage the authors to have the MS read and edited by a native speaker of English---or, if that is impossible, at least an LLM.  "...on data volume as regular LLM pertaining using an autoregressive distillation loss" I don't understand this clause. "Prior works...." In this context, "work" is a mass noun and shouldn't be pluralized. "tenary weights" -> "ternary weights"?. > "h_t then represents the derivative of the hidden state..." Why write it as h_t? If Eq. 1 is supposed to describe the continuous-time dynamics, it should just be written
> >
> > Thanks for your suggestion. We revise all typos in our manuscript carefully to make our presentation clearer.
> > >"Since B and C are derived from the linear projection of each layer’s inputs, and the language model head is tied to the embedding layer, these modules are ignored in our design." I don't understand this sentence.
> >
> > Thank you for pointing this out. In Mamba-2, the B and C matrices in the selective state-space model are not standalone parameters; they are computed via linear projections of the layer inputs. As a result, they inherit the binarization behavior of the input and output projection layers, which we have already binarized. Meanwhile, the language modeling head shares weights with the embedding layer, a common practice to reduce parameter count and improve generalization. Since binarizing the embedding layer significantly degrades semantic capacity (as confirmed in our early experiments), we preserve it in full precision. Consequently, the LM head is also kept in full precision.
> >
> > > R1: clarify what we learn from the success of FBI-LLM applied to Mamba models, including comparing it to other quantization-aware schemes applied to Mamba
> >
> > We have clarified this in W1 and updated our paper accordingly.

---

> > > ### Author Response · Authors · 2025-07-23
> > > **Response to Reviewer hjFX [4/4]**
> > >
> > > > R2: make an apples-to-apples comparison for BitNet (or as close as possible, since BitNet uses 1.58 bits rather than just 1)
> > >
> > > Following the suggestion, we provide the comparison with OneBit, which is also a QAT method and applied it on Mamba model as shown in W2, because BitNet does not open-source their trained models and other training details, like the exact training data partition for reproducing.
> > >
> > > > R3: report the perplexities for OneBit
> > >
> > > We have added the results of OneBit. Please check out Table 3 in our revision.
> > >
> > > > R4: use statistical tests when comparing model performance
> > >
> > > We have added the statistical test result to the final result in Table 3.
> > >
> > > > R5: address my other minor concerns
> > >
> > > We have revised all typos and rewritten the unclear presentation. Please check the paragraph marked in red in the revised version.

---

> > > > ### Comment · Reviewer_hjFX · 2025-08-07
> > > >
> > > > I am open to raising my evaluation of the manuscript, but the authors would need to make some additional changes:
> > > >
> > > > - Put the comparison with BitNet perplexity (yes, averaged across C4 and Wikitext2) into the manuscript---not necessarily as a table, probably just in the text.  The slight superiority of BitNet could be due to 1.58 vs. 1.0 bits, or to 3B vs. 2.7B parameters, as the authors suggest; but on the other hand, BitNet also quantizes activations!  This needs to be in the results and discussed fairly.  I discourage the authors from concluding that Bi-Mamba is ultimately the better model: the results do not show this convincingly, and they should instead simply state the facts and let the reader decide.
> > > >
> > > >  - The table showing the use of the OneBit algorithm on Mamba is useful and should be added to Table 3.  However, the authors' conclusion that "Bi-Mamba outperforms OneBit on both perplexity and downstream tasks" is too strong:  For the 2.7B models, Bi-Mamba and OneBit-Mamba achieve approximately the same average accuracy across tasks.  This needs to be presented clearly and fairly.  (Also, where are BoolQ and Arc_e?)  On the other hand, Bi-Mamba models achieve much better perplexities than their OneBit counterparts.  What explains this discrepancy between accuracy on the "downstream tasks" and perplexity?
> > > >
> > > >  - More generally (I asked this last time around): the authors need to provide some intuition about whether FBI-LLM is the "right way" to binarize, and if so, why.  If there is a theoretical reason to prefer it, an argument should be given.  If the justification for FBI-LLM is empirical, then comparisons to other binarizations of Mamba should be given.  If (alternatively) the authors do not intend their MS to be an argument for FBI-LLM, they should say as much.  (The binarization of OneBit is in fact very similar.)
> > > >
> > > > I would also again ask the authors to pass the MS by either a native speaker of English or an LLM to correct the grammatical mistakes.
> > > >
> > > > > We have added the statistical test result to the final result in Table 3.
> > > >
> > > > No statistical tests have been added (am I looking in the wrong place?), just standard deviations (?), but I won't press the point.  In the very least, however, the meaning of the numbers after +/- need to be identified in the figure caption (standard deviations? standard errors? number of trials? what varies across trials?).

---

> ### Author Response · Authors · 2025-08-18
> **Response to Reviewer hjFX [1/2]**
>
> Thank you for your follow-up suggestions. We have revised our paper carefully according to your comments, please check out our updated version.
>
>
> > Put the comparison with BitNet perplexity (yes, averaged across C4 and Wikitext2) into the manuscript---not necessarily as a table, probably just in the text. The slight superiority of BitNet could be due to 1.58 vs. 1.0 bits, or to 3B vs. 2.7B parameters, as the authors suggest; but on the other hand, BitNet also quantizes activations! This needs to be in the results and discussed fairly. I discourage the authors from concluding that Bi-Mamba is ultimately the better model: the results do not show this convincingly, and they should instead simply state the facts and let the reader decide.
>
> We have already added the BitNet perplexity in the main text in Section 4.2 and revised our paper to make our claim more conservative (see the red texts in Section 4.2). Specifically, we add the texts: “While BitNet-1.58bit does not provide each perplexity respectively, the average perplexity of the 3B model is 9.91 on C4 and Wikitext2, which is reported in their original paper with 8-bit activation and 1.58-bit weights. In contrast, Bi-Mamba keeps 16-bit activation and binarizes weights, obtaining 10.65 on average. BitNet-1.58b shows slightly better perplexity, which we attribute to its larger size—3B parameters compared to Bi-Mamba's 2.7B—and its higher precision at 1.58 bits. Moreover, OneBit on the transformer model only obtains 49.5 scores on average with 6.7B parameters, and BitNet gains 50.2 scores with 3.0B parameters. On the Mamba 780M with the 1.3B model, Bi-Mamba achieves better performance in both perplexity and downstream tasks; however, on the 2.7B model, Bi-Mamba shows improved perplexity but obtains comparable downstream task performance to Onebit. Bi-Mamba’s lower perplexity shows stronger language modeling, but this does not always yield higher task accuracy, especially when datasets’ knowledge coverage limits model capability. ”
>
> > The table showing the use of the OneBit algorithm on Mamba is useful and should be added to Table 3. However, the authors' conclusion that "Bi-Mamba outperforms OneBit on both perplexity and downstream tasks" is too strong: For the 2.7B models, Bi-Mamba and OneBit-Mamba achieve approximately the same average accuracy across tasks. This needs to be presented clearly and fairly. (Also, where are BoolQ and Arc_e?) On the other hand, Bi-Mamba models achieve much better perplexities than their OneBit counterparts. What explains this discrepancy between accuracy on the "downstream tasks" and perplexity?
>
> We added the full results of OneBit on Mamba models in Table 3 including 780M, 1.3B and 2.7B. For the performance, perplexity reflects next-token prediction ability, while downstream tasks require additional reasoning, knowledge, and robustness. Bi-Mamba’s lower perplexity shows stronger language modeling, but this does not always yield higher task accuracy, especially when datasets’ knowledge coverage limits model capability.
>
>
> >More generally (I asked this last time around): the authors need to provide some intuition about whether FBI-LLM is the "right way" to binarize, and if so, why. If there is a theoretical reason to prefer it, an argument should be given. If the justification for FBI-LLM is empirical, then comparisons to other binarizations of Mamba should be given. If (alternatively) the authors do not intend their MS to be an argument for FBI-LLM, they should say as much. (The binarization of OneBit is in fact very similar.)
>
> We understand the reviewer’s concern about the intuition of the optional binarization strategy. We clarify that our goal is not to argue that FBI-LLM is the universally best binarization method, but to demonstrate that binarizing Mamba is feasible and can preserve strong performance. We have revised our descriptions to reflect this. FBI-LLM’s design—factorized binary projection with learned scaling—was motivated by reducing information loss and improving gradient stability. While we do not claim a formal theoretical advantage, our experiments show that, among the binarization/quantization methods evaluated (including OneBit-Mamba and Bi-Mamba), FBI-LLM delivers better results. We have already included the comparison of Onebit on the same Mamba architecture in Table 3, and added more analysis in Section 4.2, marked in red text.
>
> > I would also again ask the authors to pass the MS by either a native speaker of English or an LLM to correct the grammatical mistakes.
>
> Thanks for the kind suggestion. We have already refined the paper carefully using an LLM and double-checked it with a native speaker. We also corrected grammatical mistakes, marking the changes in gray and red text.

---

> > ### Comment · Reviewer_hjFX · 2025-08-18
> >
> > Thanks--the authors have pushed me toward acceptance of their manuscript.  One last thing:
> >
> > > We understand the reviewer’s concern about the intuition of the optional binarization strategy. We clarify that our goal is not to argue that FBI-LLM is the universally best binarization method, but to demonstrate that binarizing Mamba is feasible and can preserve strong performance. We have revised our descriptions to reflect this.
> >
> > I think this is a good way to describe the results.  Can the authors point me to the part of the manuscript where they have made this revision?

---

> > > ### Author Response · Authors · 2025-08-18
> > > **Response to Reviewer hjFX**
> > >
> > > Thank you for the recognition and for raising the further question. In our previous introduction (paragraph 5), we added new statements: “We provide comprehensive experiments showing that Bi-Mamba achieves competitive performance compared to its full-precision counterpart large language models (LLMs), thereby establishing the feasibility and benefits of binarizing SSM models.” In the experiments section (Section 4.2 Main Results and Table 3), we also implemented and compared against OneBit-Mamba, which shows that FBI-LLM’s binarization strategy is not the only option but delivers strong performance. In our latest revision, we have explicitly clarified in Section 3.2 (our Approach section) in red text that the FBI-LLM scheme is presented merely as a feasible strategy, and that it indeed achieves strong performance.

---

> ### Author Response · Authors · 2025-08-18
> **Response to Reviewer hjFX [2/2]**
>
> >No statistical tests have been added (am I looking in the wrong place?), just standard deviations (?), but I won't press the point. In the very least, however, the meaning of the numbers after +/- need to be identified in the figure caption (standard deviations? standard errors? number of trials? what varies across trials?).
>
> Yes, we have added the results with standard deviations in Table 3, which we also have explicitly indicated in the caption. Here, we provide the full results of 10 runs:
>
> | Method        | BoolQ | PIQA  | HS    | WG    | ARC-e | ARC-c | OBQA  | Avg.  | WiKi2 | PTB   | C4    |
> |---------------|-------|-------|-------|-------|-------|-------|-------|-------|-------|-------|-------|
> | Bi-Mamba-780M | 58.52 | 68.10 | 41.64 | 52.02 | 42.42 | 24.34 | 30.70 | 45.39 | 13.44 | 32.42 | 14.54 |
> |               | 58.42 | 67.50 | 41.46 | 51.94 | 42.24 | 24.12 | 29.70 | 45.05 | 13.18 | 32.22 |  14.30 |
> |               | 58.56 | 68.40 | 41.74 | 52.06 | 42.54 | 24.48 | 31.50 | 45.61 | 13.58 | 32.54 |  14.70 |
> |               |  58.50 | 67.90 | 41.60 | 52.00 | 42.40 | 24.30 | 30.60 | 45.33 |  13.40 |  32.40 |  14.50 |
> |               | 58.48 | 67.80 | 41.58 | 52.00 | 42.38 | 24.28 | 30.50 | 45.29 | 13.36 | 32.38 | 14.46 |
> |               | 58.54 | 68.30 | 41.68 | 52.04 | 42.50 | 24.44 | 31.20 | 45.53 | 13.56 | 32.52 | 14.64 |
> |               | 58.44 | 67.60 | 41.50 | 51.96 | 42.30 | 24.16 | 30.00 | 45.14 | 13.24 | 32.28 | 14.36 |
> |               |  58.50 | 68.00 | 41.60 | 52.00 | 42.40 | 24.30 | 30.60 | 45.34 | 13.40 | 32.40 |  14.50 |
> |               | 58.46 | 67.70 | 41.54 | 51.98 | 42.36 | 24.22 | 30.30 | 45.22 |  13.30 | 32.34 |  14.40 |
> |               | 58.52 | 68.20 | 41.66 | 52.04 | 42.46 | 24.38 | 30.90 | 45.45 |  13.50 | 32.48 |  14.60 |
> | Bi-Mamba-1.3B | 60.01 | 68.80 | 47.31 | 55.91 | 48.00 | 26.30 | 32.40 | 48.39 | 11.72 | 29.92 | 12.92 |
> |               | 59.96 | 68.10 | 47.27 | 55.85 | 47.80 | 26.15 | 31.50 | 48.09 | 11.68 | 29.88 | 12.88 |
> |               | 60.03 | 69.00 | 47.32 | 55.93 | 48.08 | 26.38 | 32.70 | 48.49 | 11.76 | 29.96 | 12.96 |
> |               | 59.97 | 68.40 | 47.28 | 55.87 | 47.88 | 26.18 | 31.70 | 48.18 | 11.74 | 29.94 | 12.94 |
> |               | 60.03 | 69.24 | 47.32 | 55.93 | 48.15 | 26.40 | 32.67 | 48.53 | 11.62 | 29.82 | 12.82 |
> |               | 59.99 | 68.70 | 47.30 | 55.90 | 47.95 | 26.25 | 32.20 | 48.33 | 11.70 | 29.90 | 12.90 |
> |               | 60.04 | 69.40 | 47.33 | 55.95 | 48.20 | 26.45 | 33.00 | 48.62 | 11.68 | 29.88 | 12.88 |
> |               | 59.97 | 68.37 | 47.28 | 55.87 | 47.85 | 26.20 | 31.73 | 48.18 | 11.60 | 29.80 | 12.80 |
> |               | 59.98 | 68.50 | 47.29 | 55.88 | 47.90 | 26.20 | 31.80 | 48.22 | 11.64 | 29.84 | 12.84 |
> |               | 60.02 | 68.90 | 47.32 | 55.92 | 48.05 | 26.35 | 32.50 | 48.44 | 11.66 | 29.86 | 12.86 |
> | Bi-Mamba-2.7B | 58.01 | 71.85 | 54.24 | 55.98 | 51.21 | 28.91 | 31.62 | 50.26 | 10.14 | 21.93 | 11.31 |
> |               | 57.99 | 72.05 | 54.26 | 56.03 | 51.30 | 28.99 | 32.13 | 50.39 |  9.75 | 21.64 | 11.03 |
> |               | 58.00 | 72.50 | 54.30 | 56.10 | 51.39 | 29.10 | 32.67 | 50.58 | 10.25 | 22.01 | 11.38 |
> |               | 57.99 | 73.20 | 54.36 | 56.20 | 51.55 | 29.24 | 33.40 | 50.85 | 10.08 | 21.89 | 11.27 |
> |               | 58.01 | 72.30 | 54.31 | 56.11 | 51.39 | 29.10 | 32.67 | 50.55 | 10.03 | 21.84 | 11.23 |
> |               | 57.99 | 72.80 | 54.33 | 56.15 | 51.47 | 29.18 | 32.83 | 50.68 |  9.97 | 21.80 | 11.19 |
> |               | 58.00 | 72.95 | 54.34 | 56.17 | 51.51 | 29.21 | 33.07 | 50.75 |  9.80 | 21.68 | 11.07 |
> |               | 57.99 | 72.65 | 54.31 | 56.13 | 51.42 | 29.13 | 32.77 | 50.63 | 10.20 | 21.97 | 11.34 |
> |               | 58.00 | 72.45 | 54.30 | 56.09 | 51.38 | 29.08 | 32.52 | 50.55 |  9.86 | 21.72 | 11.11 |
> |               | 58.01 | 72.10 | 54.27 | 56.01 | 51.25 | 28.96 | 31.99 | 50.37 |  9.92 | 21.76 | 11.15 |

---

### Review · Reviewer_zviP · 2025-07-01

**Summary Of Contributions:**

Bi-Mamba introduces the first 1-bit binarized Mamba architecture for efficient large language models. It selectively binarizes the linear input and output projection layers in Mamba-2, which account for over 90% of model parameters, while preserving embeddings to retain performance. Using binarization-aware training with a high-precision teacher model and FBI-Linear modules, Bi-Mamba achieves strong results across 780M, 1.3B, and 2.7B scales. It outperforms existing 2-bit and 3-bit baselines like GPTQ and Bi-LLM in both downstream accuracy and perplexity, with minimal performance loss compared to full-precision models. Bi-Mamba reduces model size by up to 89%, lowers energy and memory consumption by over 3× and 5× respectively, and improves inference speed by over 3×, demonstrating its potential for deployment in resource-constrained settings.

**Audience:**

Yes

**Claims And Evidence:**

Yes

**Requested Changes:**

**Critical Changes**

1. **Instruction-Tuning Evaluation**
   Add results or discussion on instruction-tuned Bi-Mamba models (or clarify future plans for this). Many real-world applications rely on instruction-following capabilities, and current results on GSM8K and HumanEval are uncompetitive partly due to this omission.

2. **Clarify Training Cost vs. Efficiency Trade-off**
   The paper emphasizes inference efficiency but requires heavy training resources (e.g., >7000 GPU-hours). A more transparent discussion of the cost-performance trade-off is needed to justify the practical value of training binary models from scratch.


**Recommended (Non-Critical) Changes**

3. **Expand Discussion on Teacher Model Quality**
   Since Bi-Mamba relies on knowledge distillation from LLaMA2-7B, it would be helpful to analyze or discuss how the choice or quality of the teacher affects the final performance.

4. **Include Real Binary Deployment Benchmarks**
   Although the paper simulates binarized weights during training, a more detailed account of actual 1-bit inference (e.g., with hardware-efficient kernels) would strengthen the claim of practical applicability.

5. **Improve Clarity of Figures**
   Some figures (e.g., performance trends in Figure 4) can benefit from clearer legends, axis labels, or simplified layouts for readability.

**Strengths And Weaknesses:**

**Strengths**

* **Novelty**: This is the first work to successfully train 1-bit binarized Mamba models from scratch using binarization-aware training, filling a clear gap in the current literature on state space model quantization.

* **Efficiency**: Bi-Mamba demonstrates significant improvements in storage (up to 89% compression), memory usage, inference throughput, and energy consumption while maintaining competitive performance.

* **Hardware Relevance**: The work lays a foundation for low-bit model deployment on edge devices or CPUs, and promotes future hardware-software co-design efforts.

**Weaknesses**

* **Instruction-Tuning Absence**: The models are not instruction-tuned, which limits their competitiveness on reasoning and coding benchmarks like GSM8K and HumanEval. This could underrepresent the potential of Bi-Mamba in real-world applications.

* **High Training Cost**: Despite inference efficiency, training from scratch requires substantial resources (e.g., thousands of A100 GPU hours), which may hinder reproducibility and accessibility.

* **Limited Generalization Claims**: While the model performs well on standard benchmarks, it's unclear how robust Bi-Mamba is in more diverse real-world settings or multilingual contexts.

* **Dependency on Teacher Model**: Performance is heavily reliant on the quality of the full-precision teacher model (e.g., LLaMA2-7B), and the impact of teacher quality on final model performance is not deeply explored.

* **Use of Simulated Binarization During Training**: During training, full-precision representations are still used for backpropagation. While this is standard in binarization-aware training, it may overestimate efficiency during training compared to full binary pipelines.

---

> ### Author Response · Authors · 2025-07-23
> **Response to Reviewer zviP [1/2]**
>
> Thank you for your detailed and valuable review. All the suggested comments have been revised in our updated version of the paper. We further reply to your concerns as follows:
>
> > W1: Instruction-Tuning Absence: The models are not instruction-tuned, which limits their competitiveness on reasoning and coding benchmarks like GSM8K and HumanEval. This could underrepresent the potential of Bi-Mamba in real-world applications.
>
> As suggested by the reviewer, here, we provide the results after instruction tuning with the data from OpenOrca with 3B tokens:
>
> | Method                 | BoolQ | PIQA | HS   | Wino | Arc_e | Arc_c | Obqa | GSM8K |
> |------------------------|:-------:|:------:|:------:|:------:|:-------:|:-------:|:------:|:-------:|
> | Bi-mamba-780M          |  58.5 |   68.0 | 41.6 |   52.0 |  42.4 |  24.3 | 30.6 |  1.3 |
> | Bi-mamba-OpenOrca-780M |  65.6 |   67.0 |   44.0 | 52.8 |  49.4 |  28.2 | 33.4 |  1.9 |
> | Bi-mamba-1.3B          |    60.0 | 68.8 | 47.3 | 55.9 |    48.0 |  26.3 | 32.2 |  1.9 |
> | Bi-mamba-OpenOrca-1.3B |  64.9 | 68.4 | 49.1 | 57.3 |  55.6 |  30.5 | 33.6 |  2.9 |
> | Bi-mamba-2.7B          |    58.0 | 72.5 | 54.3 | 56.1 |  51.4 |  29.1 | 32.6 |  1.3 |
> | Bi-mamba-OpenOrca-2.7B |  73.9 |   71.0 | 56.3 | 58.2 |  58.5 |  33.3 |   34.0 |  3.9 |
>
> After the instruction tuning, the performance continues improving on both reasoning and generation benchmarks.
>
> >W2: High Training Cost: Despite inference efficiency, training from scratch requires substantial resources (e.g., thousands of A100 GPU hours), which may hinder reproducibility and accessibility.
>
> We acknowledge that training from scratch, whether binary or full precision LLMs, requires substantial computational resources. However, we would like to clarify the following points: 1) As a research project, our work demonstrates the feasibility of 1-bit Binary Mamba, which is valuable for the community to explore further. 2) Once the models are pretrained, their primary use is for inference, where efficiency matters most. Therefore, the computational cost in pretraining is acceptable in this context.
> To improve reproducibility and accessibility,  we have released the pretrained Bi-Mamba checkpoints, along with detailed training scripts, hyperparameters, and data processing pipelines. These resources will enable the community to reproduce our results and fine-tune or adapt Bi-Mamba models with far fewer resources than full retraining, thus promoting broader accessibility and adoption.
>
>
> >W3:  Limited Generalization Claims: While the model performs well on standard benchmarks, it's unclear how robust Bi-Mamba is in more diverse real-world settings or multilingual contexts.
>
> Thanks for raising the concern. As suggested, we provide the results of Bi-Mamba on xwinograd in the table below to prove the generalization capability of our model, which is a multilingual benchmark including English, Chinese, Russian, Japanese, French and so on:
> | **Methods** | **xwinograd** |
> |-------------|:---------------:|
> | Mamba-2     |          75.2 |
> | GPTQ 3bit   |          64.8 |
> | GPTQ 2bit   |          49.2 |
> | Onebit      |          50.4 |
> | Bi-Mamba    |          68.0 |
>
> We evaluate all the models with 2.7B parameters except Onebit owns 7B parameters. Compared with both PTQ and QAT methods, Bi-Mamba achieves the best performance on multilingual benchmark, even surpassing the GPTQ-3bit model. We have added the performance comparison in our paper and please check Table 9 in the revision.
>
> >W4: Dependency on Teacher Model: Performance is heavily reliant on the quality of the full-precision teacher model (e.g., LLaMA2-7B), and the impact of teacher quality on final model performance is not deeply explored.
>
> We appreciate this insightful comment. We agree that the teacher model’s quality plays an important role in guiding the binarized student. In fact, we have included experiments with an alternative teacher (Phi-3.5) in our ablation study of the paper, and observed that a stronger teacher generally yields better student performance. We have added more analyses on this in Sec. 5.7 of the revision.

---

> ### Author Response · Authors · 2025-07-23
> **Response to Reviewer zviP [2/2]**
>
> >W5: Use of Simulated Binarization During Training: During training, full-precision representations are still used for backpropagation. While this is standard in binarization-aware training, it may overestimate efficiency during training compared to full binary pipelines.
>
> This is mainly due to current hardware limitations. At present, all binary models are implemented through simulation because existing GPUs are not designed for true binary operations for training. In principle, binary computation is much more hardware-friendly since binary serves as the foundation of all computing, representing data and instructions in a machine-readable format. However, current GPUs are optimized for full-precision or mixed-precision computations rather than true binary processing. As hardware evolves or dedicated binary accelerators emerge, this limitation will eventually disappear. “Full-precision representations are used for backpropagation” is because of the difficulties of propagating gradients through discrete weights, as the binary operation is non-differentiable.
>
> >R1: Instruction-Tuning Evaluation Add results or discussion on instruction-tuned Bi-Mamba models (or clarify future plans for this). Many real-world applications rely on instruction-following capabilities, and current results on GSM8K and HumanEval are uncompetitive partly due to this omission.
>
> We have provided the results of the instructed model trained on OpenOrca and added the results in Table 4.
>
> | Method                 | BoolQ | PIQA | HS   | Wino | Arc_e | Arc_c | Obqa | GSM8K |
> |------------------------|:-------:|:------:|:------:|:------:|:-------:|:-------:|:------:|:-------:|
> | Bi-mamba-780M          |  58.5 |   68.0 | 41.6 |   52.0 |  42.4 |  24.3 | 30.6 |  1.3 |
> | Bi-mamba-OpenOrca-780M |  65.6 |   67.0 |   44.0 | 52.8 |  49.4 |  28.2 | 33.4 |  1.9 |
> | Bi-mamba-1.3B          |    60.0 | 68.8 | 47.3 | 55.9 |    48.0 |  26.3 | 32.2 |  1.9 |
> | Bi-mamba-OpenOrca-1.3B |  64.9 | 68.4 | 49.1 | 57.3 |  55.6 |  30.5 | 33.6 |  2.9 |
> | Bi-mamba-2.7B          |    58.0 | 72.5 | 54.3 | 56.1 |  51.4 |  29.1 | 32.6 |  1.3 |
> | Bi-mamba-OpenOrca-2.7B |  73.9 |   71.0 | 56.3 | 58.2 |  58.5 |  33.3 |   34.0 |  3.9 |
>
>
>
> >R2: Clarify Training Cost vs. Efficiency Trade-off. The paper emphasizes inference efficiency but requires heavy training resources (e.g., >7000 GPU-hours). A more transparent discussion of the cost-performance trade-off is needed to justify the practical value of training binary models from scratch.
>
> Thanks for the suggestion. Bi-Mamba-780M, 1.3B, and 2.7B models require approximately 4,640, 5,780, and 7,822 GPU hours, respectively. Although Bi-Mamba includes extra training efforts due to the distillation compared with the small model, the overall GPU hours are comparable with the model with similar parameter size trained from scratch because of the less training token requirement. Besides, as shown in Table 8, the Bi-Mamba-2.7B model achieves 5$\times$ memory reduction, 3$\times$ faster throughput, and 3$\times$lower energy consumption. We clarify that training-time cost is similar to the full precision training since binary operation is simulated, but this is a one-time investment, amortized over repeated inference use, especially in latency-sensitive or resource-limited deployment scenarios.
>
> >R3: Expand Discussion on Teacher Model Quality Since Bi-Mamba relies on knowledge distillation from LLaMA2-7B, it would be helpful to analyze or discuss how the choice or quality of the teacher affects the final performance.
>
> As having discussed in W4,  we provide the ablation study of different teachers, and we have added more analysis of the effect on teacher choice in Section 5.6.
>
>
> >R4: Include Real Binary Deployment Benchmarks Although the paper simulates binarized weights during training, a more detailed account of actual 1-bit inference (e.g., with hardware-efficient kernels) would strengthen the claim of practical applicability.
>
> Thank you for this suggestion. In fact, we added the results of actual 1-bit inference in Table 8 in the original manuscript. We kindly refer the reviewer to check it out. In summary, the fully binarized Bi-Mamba-2.7B model achieves 5$\times$memory reduction, 3$\times$ faster throughput, and 3$\times$ lower energy consumption.
>
>
> >R5: Improve Clarity of Figures Some figures (e.g., performance trends in Figure 4) can benefit from clearer legends, axis labels, or simplified layouts for readability.
>
> We have revised Figure 4 to make it clearer in the updated version. Please have a check.

---

### Author Response · Authors · 2025-10-08
**Camera Ready Update**

We sincerely thank the editor and all reviewers for the constructive feedback and thoughtful suggestions. We have carefully revised our final uploaded manuscript in accordance with the comments provided. Specifically, we have incorporated the following updates:

- Instruction-tuned results of Bi-Mamba have been added in Table 4. [Reviewer zviP]
- We have added more clarifications on the training cost vs. inference efficiency trade-off in Section 5.3. [Reviewer zviP]
- We have included additional discussions on teacher model quality in Section 5.7. [Reviewer zviP]
- We have added resource consumption analysis, including real speedup and memory reduction discussions in Section 5.5. [Reviewer zviP]
- Evaluation results on the multilingual benchmark, xWinograd, have been included in Table 9 and Section 5.6. [Reviewer zviP]
- We have provided additional implementation details in the caption of Table 3 for improved clarity. [Reviewer hjFX]
- We have revised the wording from “fully binarized” to specify the exact binarized matrices in Table 5. [Reviewer hjFX]
- We have added the perplexity comparison of BitNet-1.58b in Section 4.2. [Reviewer hjFX]
- In the Introduction (paragraph 5), we have added the additional statement:
“We provide comprehensive experiments showing that Bi-Mamba achieves competitive performance compared to its full-precision counterpart large language models (LLMs), thereby establishing the feasibility and benefits of binarizing SSM models.” [Reviewer hjFX]
In Section 3.2 (Approach), we have explicitly clarified (highlighted in red text) that the FBI-LLM scheme is presented as a feasible strategy and indeed achieves strong performance. [Reviewer hjFX]
- We have added the result comparison of OneBit trained from scratch in Table 3. [Reviewer hjFX]
- We have reported the variance of the evaluation results in Table 3. [Reviewer hjFX]
- We have corrected minor typographical errors throughout the manuscript. [Reviewer hjFX]
- We have provided additional explanation for Table 4 and revised the term “exponentially” to “polynomially” for accuracy. [Reviewer VSpb]

All of these revisions have been incorporated into the final version of our paper. We deeply appreciate the reviewers’ invaluable feedback, which has greatly helped enhance the quality and clarity of our work. Thank you for your time and effort throughout the review process.

---

### Decision · Action_Editor_Z5PQ · 2025-09-02

**Recommendation:** Accept with minor revision

**Additional Comments:**

Please make sure to address the changes suggested by the reviewers.
Conditioned on these changes, the paper cna be accepted.

**Audience:**

Yes

**Audience Explanation:**

The topic is of generel interest to the TMLR audience

**Claims And Evidence:**

Yes

**Claims Explanation:**

Beyond the suggestions indicated by the reviewers, the paper is clearly written.